# Latent Feature Alignment: Discovering Biased and Interpretable Subpopulations in Face Recognition Models

## Abstract

Modern face recognition models achieve high overall accuracy but continue to exhibit systematic biases that disproportionately affect certain subpopulations. Conventional bias evaluation frameworks rely on labeled attributes to form subpopulations, which are expensive to obtain and limited to predefined categories. We introduce Latent Feature Alignment (LFA), an attribute-label-free algorithm that uses latent directions to identify subpopulations. This yields two main benefits over standard clustering: (i) semantically coherent grouping, where faces sharing common attributes are grouped together more reliably than by proximity-based methods, and (ii) discovery of interpretable directions, which correspond to semantic attributes such as age, ethnicity, or attire. Across four state-of-the-art recognition models (ArcFace, CosFace, ElasticFace, PartialFC) and two benchmarks (RFW, CelebA), LFA consistently outperforms k-means and nearest-neighbor search in intra-group semantic coherence, while uncovering interpretable latent directions aligned with demographic and contextual attributes. These results position LFA as a practical method for representation auditing of face recognition models, enabling practitioners to identify and interpret biased subpopulations without predefined attribute annotations.

## 1 Introduction

Face recognition technology (FRT) has seen widespread adoption in applications ranging from unlocking smartphones and securing sensitive facilities to assisting law enforcement and aiding in search operations. Rapid advancements in FRT have driven accuracy to levels that exceed human performance (Deng et al., 2019; Liu et al., 2017; Phillips et al., 2018; Schroff et al., 2015), promising enhanced efficiency and error reduction. Yet, as these systems proliferate, so do the risks associated with their misuse—ranging from privacy breaches to civil rights violations (Castelvecchi, 2020; Van Noorden, 2020; U.S. Commission on Civil Rights, 2024). Numerous studies have highlighted systematic biases in these systems (Cavazos et al., 2020; Drozdowski et al., 2020; Garcia et al., 2019; Wang & Deng, 2020; Gong et al., 2021; Robinson et al., 2020; Serna et al., 2022; Terhörst et al., 2021; Yucer et al., 2024). These biases can result in unfair treatment, exacerbating societal inequalities and undermining trust in their use (Garvie et al., 2016; Clayton, 2024).

Bias in face recognition algorithms refers to systematic differences in performance that affect certain groups of people based on characteristics such as race, gender, age, or other demographic factors. These biases creep in at various stages of the algorithm's lifecycle, from data collection and preprocessing to model design.

Traditionally, bias identification and mitigation rely on supervised approaches, where labeled datasets are used to define and evaluate group-specific performance gaps. However, such methods depend on explicit annotations (Terhörst et al., 2021), which may be unavailable, costly, or prone to unintended biases introduced by human annotators and unnoticed confounders (Balakrishnan et al., 2021). While synthetic datasets have been proposed as an alternative (Liang et al., 2023), they still only account for predefined attributes, leaving room for unconsidered biases to persist.

The challenge of identifying and analyzing bias in face recognition models has thus been largely constrained to known, predefined attributes. We address this limitation by developing a *label-free*

method that autonomously uncovers subpopulations correlated with performance disparities, without requiring attribute annotations or synthetic manipulations.

We introduce **Latent Feature Alignment (LFA)**, an algorithm that uses latent directions in the embedding space to discover subpopulations and reveal interpretable structure in face recognition models. Using latent directions provides two key benefits over standard clustering: (i) **semantically coherent grouping**, where samples sharing common attributes are grouped more reliably than by distance-based methods, and (ii) **discovery of interpretable directions**, which align with semantic attributes such as age, ethnicity, or attire and expose bias-related subpopulations.

Our contributions are:

- **Latent Feature Alignment (LFA):** A novel attribute-label-free algorithm that discovers semantically coherent subpopulations in face recognition embeddings by aligning samples along latent directions.

- **Semantic coherence:** LFA consistently outperforms nearest-neighbor search and $k$-means clustering in grouping semantically related samples across multiple datasets (RFW, CelebA) and models (ArcFace, CosFace, ElasticFace, PartialFC).

- **Interpretable latent directions:** Directions discovered in the latent space by LFA correspond to human-interpretable attributes, validated both quantitatively and qualitatively.

- **Bias discovery without annotations:** LFA identifies subpopulations that exhibit systematic disparities in recognition performance, without requiring attribute labels.

- **Resources:** We validate vision–language model annotations as a reliable large-scale evaluation tool and release attribute annotations for RFW to support future bias auditing research.

## 2 RELATED WORK

Our work intersects two central areas of research: (1) the discovery and interpretation of latent representations in computer vision models, and (2) the detection of bias in deep learning models. Prior work in these domains can be broadly classified by their supervision paradigm (supervised vs. unsupervised) and their application focus (interpretability or bias mitigation).

### 2.1 SUPERVISED APPROACHES

**Latent Concept Discovery in Vision Models.** Early efforts to interpret neural networks revealed that human-defined concepts can often be represented as directions in latent space rather than as isolated neurons. Notably, *Concept Activation Vectors* (CAVs) (Kim et al., 2018) are learned by training a linear classifier to distinguish between concept-specific activations and random activations. Subsequent work has extended this idea to semi-supervised settings, using latent space decomposition to uncover concept vectors with partial labels (Graziani et al., 2023; O'Mahony et al., 2023).

**Interpretable Feature Directions in Face Recognition.** Plesh et al. (2024) explored interpretability in face recognition by identifying semantically meaningful directions in embedding spaces. Their method relied on annotated facial attributes and blended-image techniques to recover latent directions, which were then traversed for qualitative interpretation.

**Synthetic Image Generation for Bias Detection.** Counterfactual data augmentation has been proposed to diagnose bias by generating synthetic faces that selectively vary target attributes (Balakrishnan et al., 2021; Hutchinson et al., 2019; Joo & Kärkkäinen, 2020; Liang et al., 2023). While these methods support causal analyses, they are restricted to attributes chosen a priori, limiting their ability to uncover unexpected sources of bias.

### 2.2 UNSUPERVISED APPROACHES

**Discovery of Latent Semantics in Generative Models.** Advances in Generative Adversarial Networks (GANs) have significantly enhanced our ability to interpret and manipulate latent spaces, leading to controlled and meaningful image generation (Härkönen et al., 2020). For example, SeFa

(Shen & Zhou, 2021) introduced a closed-form factorization for uncovering latent semantic directions without additional training or sampling. Other approaches enforce orthogonality and distinctiveness of latent directions (Song et al., 2023) or use contrastive objectives (Yüksel et al., 2021). And others learn clusters corresponding to meaningful attributes (Georgopoulos et al., 2022). These approaches demonstrate the promise of latent direction analysis, though they are generally applied to generative rather than discriminative models.

**Unsupervised Bias Discovery via Clustering.** Clustering-based methods have been proposed to detect bias in the absence of labels. For example, Krishnakumar et al. (2021) used hierarchical clustering of embeddings to identify subgroups with degraded classification performance in models trained for image classification. Glüge et al. (2020) showed that simple clustering-based measures are not reliable indicators of bias in face recognition models, arguing for more nuanced approaches.

These findings highlight both the promise and the limitations of unsupervised grouping for bias auditing: while clustering can reveal potential subgroups of concern, naïve distance-based methods may fail to align with semantically meaningful attributes or with actual performance disparities. Our method addresses these challenges by introducing *Latent Feature Alignment (LFA)*, which forms groups along latent directions in the embedding space. This provides higher semantic coherence than local similarity-based clustering, while simultaneously uncovering interpretable directions that correspond to meaningful, previously unlabeled factors of variation.

## 3 METHODOLOGY

### 3.1 LATENT FEATURE ALIGNMENT (LFA)

This work presents a novel framework that operates in the embedding space of face recognition models to discover subpopulations with distinct interpretable characteristics useful for bias auditing. At the core of our approach lies *Latent Feature Alignment (LFA)*, an algorithm that uses latent directions to identify semantically similar face images.

The algorithm computes a latent direction $\vec{v}$ from a group of image embeddings. Specifically, the latent direction is obtained by taking the normalized average of the group's embeddings (see Algorithm 1). Next, among all face embeddings in the dataset, the algorithm identifies the one most aligned with this latent direction and adds it to the group (see Algorithm 2). Alignment is measured by projecting each embedding onto $\vec{v}$ and selecting the embedding with the highest projection value.

This process repeats until the projection of the most aligned face embedding falls below a predefined threshold ($\tau$). By projecting the embeddings onto the latent vector, we capture the extent to which that direction is represented in each image. Note that all embeddings are normalized, so the embedding with the largest projection corresponds to the one with the greatest component of $\vec{v}$ in its direction.

We now present the mathematical formulation of the method. Let $\mathcal{X} = \{x_i\}_{i=1}^{N}$ be a dataset of $N$ face images from which we aim extract a semantically consistent group. Using a deep feature extractor $\phi(\cdot)$, each image is mapped to a face embedding:

$$x_i \mapsto \phi(x_i) = \vec{e}_i \in \mathbb{R}^d$$

where $d$ denotes the dimensionality of the embedding space. Let $\ell_i \in \{1, \ldots, C_{\max}\}$ denote the identity label of $x_i$.

Given a current subset of indices $\mathcal{S} \subset \{1, \ldots, N\}$ of size $n$, define the count of samples with identity $\ell$ and the number of unique identities in $\mathcal{S}$ as:

$$c_\ell = |\{i \in \mathcal{S} : \ell_i = \ell\}|, \qquad C = |\{\ell_i : i \in \mathcal{S}\}|$$

The algorithm then iteratively performs the following steps:

1. Compute the latent direction of the subset:

$$\vec{v} = \frac{1}{C} \sum_{j \in \mathcal{S}} w_j \vec{e}_j \qquad \text{with} \qquad w_j = \frac{1}{c_{\ell_j}} \quad \text{(weight of sample } j\text{)} \tag{1}$$

---

**Algorithm 1** GETLATENTDIRECTION

---

**Require:** Embeddings $\{\vec{e}_j\}_{j=1}^n$ and identity labels $\{\ell_j\}_{j=1}^n$
**Ensure:** Latent direction $\vec{v}$
1: $\mathcal{I} \leftarrow \{\ell_j : j = 1, \ldots, n\}$          ▷ Set of unique identities
2: **for** each identity $i \in \mathcal{I}$ **do**
3:     $c_i \leftarrow |\{j : \ell_j = i\}|$          ▷ Count samples per identity
4: **end for**
5: **for** $j = 1, \ldots, n$ **do**
6:     $w_j \leftarrow 1/c_{\ell_j}$          ▷ Inverse frequency weighting
7: **end for**
8: $\vec{v} \leftarrow \sum_{j=1}^n w_j\,\vec{e}_j$          ▷ Weighted sum
9: **return** $\vec{v}$

---

**Algorithm 2** LATENT FEATURE ALIGNMENT

---

**Require:** Dataset embeddings $E$, initial group $S$
1: **while** $\max\{p\} \geq \tau$ **do**
2:     $\vec{v} \leftarrow$ GETLATENTDIRECTION($S$)
3:     $p \leftarrow E \cdot \vec{v} \,/\, \|\vec{v}\|_2$          ▷ Project embeddings onto direction
4:     $i \leftarrow \arg\max\{p\}$          ▷ Most aligned embedding
5:     $S \leftarrow S \cup \{\vec{e}_i\}$          ▷ Add to group
6:     $E \leftarrow E \setminus \{\vec{e}_i\}$          ▷ Remove
7: **end while**
8: **return** $S, \vec{v}$

---

2. Find the index of the most aligned embedding not already in $\mathcal{S}$:

$$i* = \underset{k \notin \mathcal{S}}{\arg\max} \frac{\langle \vec{e}_k, \vec{v} \rangle}{\|\vec{v}\|_2} \tag{2}$$

3. Check the stopping condition:

$$\frac{\langle \vec{e}_{i*}, \vec{v} \rangle}{\|\vec{v}\|_2} < \tau \tag{3}$$

4. Update the subset by adding the index:

$$\mathcal{S} = \mathcal{S} \cup \{i*\} \tag{4}$$

## 3.2 AUTOMATED DISCOVERY OF INTERPRETABLE AND BIASED GROUPS

*Latent Feature Alignment* relies on the existence of a coherent underlying factor shared across a group of images. When such a factor is present, the latent direction aggregates and reflects the common representation, yielding a semantically interpretable vector. Conversely, if the images are unrelated, the latent direction captures inconsistent signals, resulting in noise and an incoherent mixture of faces without a distinct representation. In this case, projections become uninformative, and the resulting groups lose interpretability.

The effectiveness of the method therefore depends critically on the initial group of images. Since no prior knowledge about groups is assumed, two strategies are possible: (1) exhaustive enumeration of small subsets (e.g., groups of size $k < 10$), or (2) initialization through pre-clustering. We adopt the latter, which scales to large datasets and provides semantically consistent starting points for *LFA*. Specifically, we construct an undirected similarity graph where each node corresponds to an embedding and edges are formed between embeddings exceeding a cosine similarity threshold of 0.5. Each connected component defines an initial group, while isolated nodes form singleton groups.

To compute the latent direction of a group, we average the embeddings. However, datasets often contain unbalanced numbers of images per identity, which can bias the mean toward dominant identities. To mitigate this, we use a weighted average that equalizes the contribution of each identity, ensuring that the latent direction reflects group-level semantic factors rather than identity frequency (Algorithm 1).

Table 1: Semantic coherence (↓ average attribute distance) across CelebA and RFW datasets. *LFA* thresholds and $k$ values were chosen to yield comparable group sizes ($N \approx 100$). Across both datasets, *LFA* is expected to produce more semantically coherent groups than *nearest-neighbor search* (NNS), *k-means*, and *spherical k-means*.

| Method | CelebA | | | | RFW | | | |
|---|---|---|---|---|---|---|---|---|
| | ArcFace | CosFace | ElasticFace | PartialFC | ArcFace | CosFace | ElasticFace | PartialFC |
| NNS | 8.61 ± 0.02 | 8.59 ± 0.02 | 8.60 ± 0.02 | 8.52 ± 0.02 | 3.12 ± 0.01 | 3.04 ± 0.01 | 3.13 ± 0.01 | 3.10 ± 0.01 |
| k-means | 7.73 ± 0.06 | 7.73 ± 0.06 | 7.66 ± 0.06 | 7.61 ± 0.06 | 3.01 ± 0.06 | 2.81 ± 0.07 | 2.96 ± 0.07 | 3.01 ± 0.06 |
| Spherical k-means | 8.06 ± 0.05 | 8.07 ± 0.04 | 8.04 ± 0.05 | 7.97 ± 0.05 | 2.65 ± 0.08 | 2.38 ± 0.08 | 2.63 ± 0.08 | — |
| DBSCAN | 6.94 ± 0.17 | 7.17 ± 0.14 | 7.47 ± 0.10 | 7.50 ± 0.19 | 2.13 ± 0.30 | 2.29 ± 0.30 | 2.46 ± 0.43 | — |
| *LFA (ours)* | **6.38 ± 0.02** | **6.37 ± 0.02** | **6.53 ± 0.02** | **6.25 ± 0.03** | **1.86 ± 0.02** | **1.52 ± 0.02** | **1.84 ± 0.02** | **1.54 ± 0.03** |

After discovering semantically coherent groups via *LFA*, we evaluate their bias characteristics by analyzing impostor similarity scores within each group. Specifically, we compute the cosine similarity between embeddings belonging to different identities within the group. A high average impostor similarity indicates that the group is a biased subpopulation: the impostor score distribution shifts rightward, leading to elevated error rates in face recognition systems (Serna et al., 2022).

## 4 EXPERIMENTAL EVALUATION

We assess the effectiveness of our proposed method, Latent Feature Alignment (*LFA*), in identifying semantically meaningful and potentially biased subpopulations within face recognition datasets. Our evaluation focuses on three key aspects: (1) the semantic coherence of discovered clusters, (2) the interpretability of latent feature directions, and (3) the relationship between discovered groups and recognition performance bias. We conduct experiments on two widely-used face datasets: Racial Faces in the Wild (RFW) (Wang et al., 2019) and CelebA (Liu et al., 2015), and evaluate four state-of-the-art face recognition models.

### 4.1 DATASETS AND MODELS

We conduct our experiments on the following datasets:

**RFW** (Wang et al., 2019): A benchmark designed to measure racial bias in face recognition systems. It contains four subsets—Caucasian, African, Asian, and Indian—each with approximately 3,000 identities and 10,000 images. Its primarily source was the MS-Celeb-1M Guo et al. (2016) dataset.

**CelebA** (Liu et al., 2015): A large-scale dataset consisting of over 200,000 images of more than 10,000 celebrities, annotated with 40 binary attributes. It serves as a benchmark for attribute-based classification.

We annotate and validate RFW using state-of-the-art open-source Vision-Language Models (VLMs), including InternVL3 (Chen et al., 2024), Ola (Liu et al., 2025), Ovis2 (Lu et al., 2024), Qwen2.5-VL (Bai et al., 2025), and SAIL-VL (Dong et al., 2025). These annotations span 10 demographic and appearance attributes, enabling quantitative evaluation of the semantic coherence of discovered clusters. (Details of the annotation and validation in Appendix A.)

For our face embeddings, we use four widely adopted face recognition models: ArcFace (Deng et al., 2019), CosFace (Wang et al., 2018), Partial FC (An et al., 2021), and ElasticFace (Boutros et al., 2022).

### 4.2 SEMANTIC COHERENCE OF GROUPS

We measure intra-group attribute consistency using average Hamming distance between binary facial attributes (lower is better), which reflects the semantic homogeneity of each group. This metric quantifies the number of attribute differences between all pairs of images in the group, with the result representing the average number of changing attributes.

We compare against four clustering algorithms *k-means*, *spherical k-means*, *Nearest Neighbor Search*, and *DBSCAN*. Unlike classical clustering, we do not optimize for total coverage or disjointness; groups may overlap, and low-quality or ambiguous samples may remain ungrouped. To

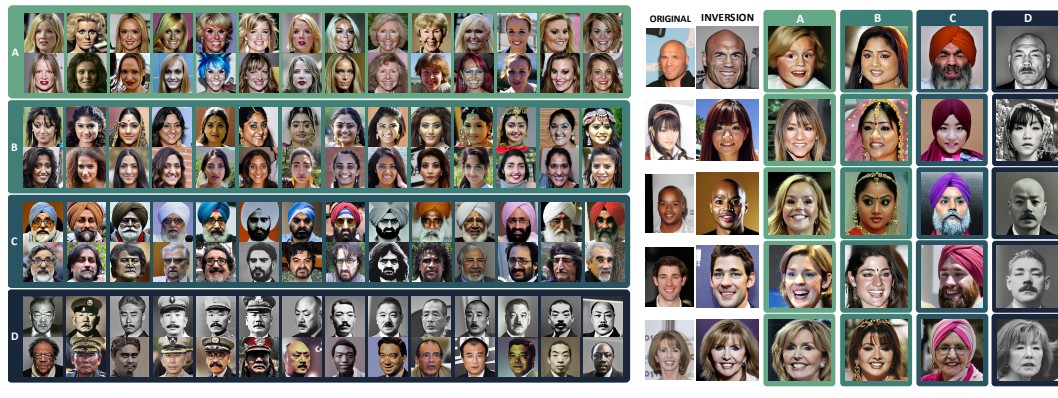

(a) Removal                                               (b) Addition

Figure 1: **Left: Removal of semantic attributes from face images**. For each group (A, B, C and D) the top image shows the inversion (embedding decoded) of the original image and the bottom one the original image after traversing back the embedding along the latent direction of the group. **Right: Addition of semantic attributes**. First column shows the original image, second column the inversion, and the rest of the columns (A-D) the original image decoded after traversing it along the latent direction of groups A-D.

ensure fairness, we match the average group size between LFA and k-means: we run LFA with a threshold that yields groups of approximately $n$ samples and set every hyperparameter of the algorithms such that the average cluster size is also $\sim n$. We note that optimizing $k, \epsilon$ with standard clustering indices (e.g., Silhouette, Calinski-Harabasz) recovers the number of identities, which is unsuitable for our goal of discovering semantic attributes rather than identity partitions. This way, density-based spatial clustering of applications with noise (*DBSCAN*), *k-means*, and *spherical k-means* produce disjoint clusters with significant variation in size (groups of 2 samples and groups of $> 10,000$ samples). We also compare against *Nearest Neighbor Search (NNS)*, that is similar to LFA in the sense that it forms groups that can overlap and are not necessarily disjoint; for fairness we set k to $n$. This way, all compared algorithms posses an average group size of $n$.

Table 1 shows the comparison for $n = 100$. In the RFW dataset the average distance for standard clustering is a around 3 attributes when using all models, while for our algorithm is almost half. In the CelebA dataset our algorithm yields also a better semantic clustering. The reason behind the CelebA results being more than two times worse is twofold, first CelebA has 40 binary attributes in contrast to the 10 classes of the RFW annotations, and second, it has significant label noise Hand et al. (2018); Lingenfelter et al. (2022); Wu et al. (2023). (see Appendix A for details on robust annotation and validation, and Appendix E for quantitative ablations.)

We performed hyperparameter ablation experiments for all baseline algorithms and *LFA* to facilitate a meaningful comparison. The results include the average semantic coherence (attribute-distance) for the top-100 groups, with confidence intervals, and indicate similar performance trends. This comparison allows us to assess the relative strengths and weaknesses of each method, highlighting how hyperparameter adjustments impact semantic coherence. (Appendix Figures 11–16.)

We observe that, in contrast to the standard clustering baselines, *LFA* groups tend to be more cohesive and semantically similar. The advantage stems from *LFA*'s ability to exploit the structure of the latent space, rather than relying purely on distance-based procedures that cannot capture broader semantic coherence.

### 4.3 INTERPRETABILITY OF LATENT DIRECTIONS

We evaluate qualitatively and quantitatively whether latent directions discovered by LFA correspond to meaningful attributes. For that, we decode embeddings as they are traversed along each latent direction. To decode an embedding we use the identity-preserving generative model arc2face Papantoniou et al. (2024), that generates high-quality images of any subject given only its ArcFace embedding.

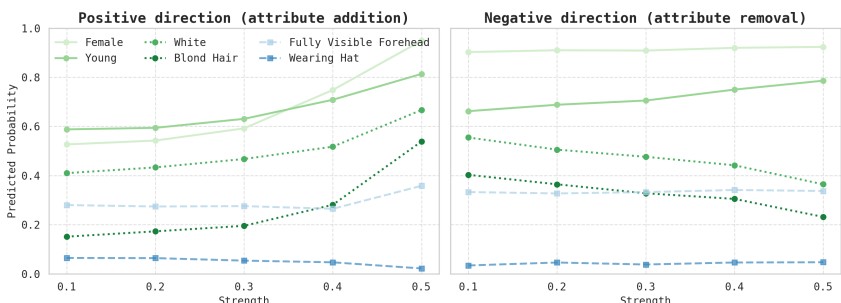

Figure 2: **Attribute probability vs traversal strength**. Predicted probabilities from an attribute classifier for images traversed along the latent direction of Group A (young, white, blonde, female). Relevant attributes (e.g., *Blond Hair, Young, Female, White*) change monotonically with traversal strength, while unrelated attributes (e.g., *Wearing Hat, Fully Visible Forehead*) remain stable.

We selected four groups formed with our method in RFW database and tested the latent direction of each one. We traversed the directions to add or remove the common attributes. We removed the attributes of the face images in the selected groups by moving along its negative direction, and we added the attributes to random face images from CelebA dataset, similarly moving along the positive direction. The qualitative visual results are shown in Figures 1a and 1b. They reveal that latent directions discovered by *LFA* correspond to interpretable concepts such as blonde female, Indian female, turban, or gray images of Chinese men. The group A is composed of white blonde females, group B are young Indian women, group C Indian males with turbans, and group D Chinese men with mustaches in black-and-white.(Appendix D provides additional traversal visualizations and examples across multiple groups.)

Note that the directions were discovered using images from one database (RFW) but applied to the images of another database (CelebA), we did this to demonstrate the latent direction is a property of the latent space produced by the model and not tied to a particular database.

Also, note that traversing any direction in an arbitrary manner results in the decoding of a meaningless or generic image that has nothing to do with the original identity. The results are surprising since our algorithm operates on discriminative face recognition embeddings (e.g., ArcFace, CosFace), which are optimized for verification accuracy and not trained for semantic editing like generative models.

It can be seen that the traverse does not work equally well for every image or identity. This can be especially seen in group D, where some pictures has the mustache removed and others do not. Two key factors were identified as determinants: one, the image quality, and two, the identity. (More details on how traverse the embedding space in Appendix C.)

To quantify whether dominant directions discovered by LFA correspond to meaningful attributes we train an attribute classifier on CelebA following Lingenfelter & Hand (2021) and measure predicted attribute probabilities as embeddings are traversed and decoded.

Like for the qualitative analysis we used the latent directions identified by LFA on RFW onto CelebA embeddings. We traversed embeddings in both the positive (attribute addition on 1,000 random CelebA samples) and negative (attribute removal; 100 RFW samples aligned with the group's attributes) directions, measuring the predicted probabilities of multiple facial attributes at different interpolation strengths.

Figure 2 reports results for Group A (*young, white, blonde, female*). Traversals along the latent direction reveal meaningful and monotonic changes in relevant attributes (e.g., *Blond Hair, Young, Female, White*) while unrelated attributes (e.g., *Wearing Hat, Fully Visible Forehead*) remain stable.

Interestingly, we also observe some exceptions. For example, *Young* increases in both positive and negative traversals, and *Female* remains flat in the negative direction despite increasing in the positive one. This is consistent with what we observe in the qualitative results: sometimes, performing addition affects gender, but removal does not. Specifically, latent directions of female groups (A

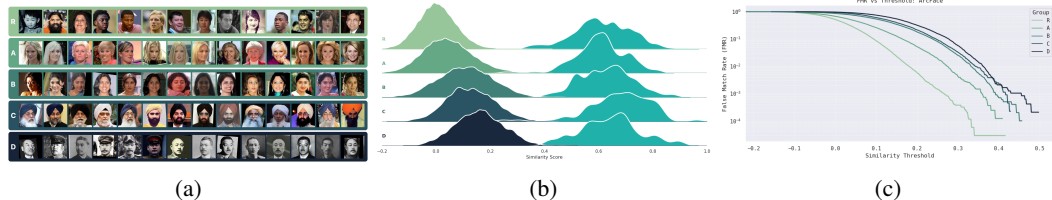

(a)  (b)  (c)

Figure 3: *Qualitative and quantitative evidence of bias discovered by LFA on ArcFace.* **Left:** Example images from different groups (R: random sample, A: Caucasian blonde females, B: young Indian women, C: Indian males with turbans, D: Chinese men with mustaches in black-and-white photographs). **Center:** Similarity score distributions for groups R–D. The right distribution of each group corresponds to genuine pairs (higher similarity), while the left one represents impostor pairs (lower similarity). Darker colors indicate higher error rates. **Right:** False Match Rate (FMR) curves for groups R–D, computed from genuine and impostor distributions (center). The extent of bias is indicated by the hue of the color used; darker hues denote a higher degree of bias.

Table 2: Performance comparison of models under different training schemes. Comparison of False Match Rate (FMR) at fixed threshold (t = 0.2) between our method and attribute annotations. The four models are evaluated on four discovered groups plus a random group. Each subset includes the number of images and comparisons.

| Model | Random | A (130 / 8k) | | B (143 / 10k) | | C (96 / 4k) | | D (100 / 5k) | |
|---|---|---|---|---|---|---|---|---|---|
| | (265/35k) | LFA | Annotated | LFA | Annotated | LFA | Annotated | LFA | Annotated |
| ArcFace | 0.0057 | 0.052 (x9) | 0.0156 (x3) | 0.180 (x32) | 0.052 (x9) | 0.215 (x38) | 0.152 (x27) | 0.318 (x56) | 0.145 (x25) |
| CosFace | 0.0045 | 0.052 (x12) | 0.0196 (x4) | 0.245 (x54) | 0.068 (x15) | 0.196 (x44) | 0.138 (x31) | 0.279 (x62) | 0.117 (x26) |
| ElasticFace | 0.0043 | 0.057 (x13) | 0.0196 (x5) | 0.167 (x39) | 0.052 (x12) | 0.285 (x66) | 0.188 (x44) | 0.436 (x101) | 0.175 (x41) |
| PartialFC | 0.0016 | 0.016 (x10) | 0.0057 (x4) | 0.106 (x66) | 0.024 (x15) | 0.081 (x51) | 0.064 (x40) | 0.140 (x88) | 0.053 (x33) |

and B) change the gender of males to female, but latent directions of male groups (C and D) do not change it for females 1b.

While we do not yet have a definitive explanation, these effects may reflect non-linear structure in the embedding space, an underfitted representation, or entanglement of certain attributes. Such observations highlight both the interpretability and the complexity of the latent space learned by face recognition models.

Importantly, these behaviors are not specific to Arc2Face. In Appendix C (Decoder-Agnostic Traversals), we repeat the same positive/negative traversals using InfiniteYou (Jiang et al., 2025), which differs from Arc2Face in both the face encoder (IR-SE50 vs. ArcFace-R100) and the decoder (Flux vs. Stable Diffusion). The resulting attribute-probability curves show the same monotonic changes in the group-defining attributes, supporting our interpretation.

### 4.4 BIAS DISCOVERY WITH ATTRIBUTE ANNOTATIONS

We evaluate whether the subpopulations discovered by LFA correspond to groups with higher error rates, and whether these trends are consistent with groups defined using explicit attribute annotations. We use the RFW database given that was specifically designed to measure bias (race bias).

Figure 3a displays example faces for different groups (R: random sample, A: Caucasian blonde females, B: young Indian women, C: Indian males with turbans, D: Chinese men with mustaches in black-and-white photographs). These exemplars illustrate the semantic coherence discovered by our method.

Figure 3b shows the distribution of cosine-similarity scores for genuine and imposter pairs within each group for the Arcface model. Genuine pairs are pairs of images belonging to the same person and imposter pairs are formed by pairing images from different identities. Similarity scores are computed using the cosine similarity of the corresponding face embeddings. Higher similarity values correspond to genuine pairs (right distribution of each group), while lower values correspond to imposters (left distribution).

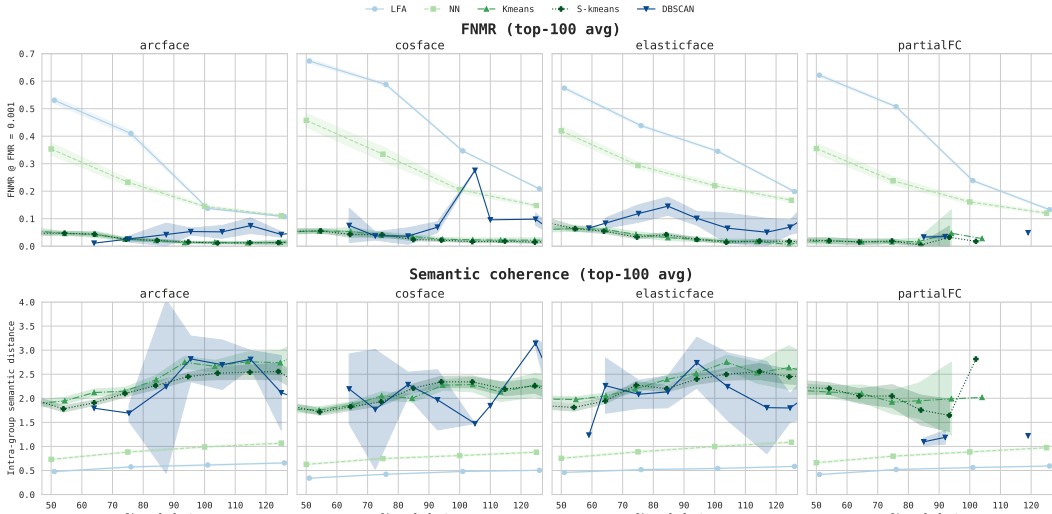

Figure 4: **Performance comparison of LFA and baseline algorithms (*Nearest Neighbors*, *k-means*, *spherical k-means*, *DBSCAN*) across four FR models.** For each method and each FR backbone, we aggregate all discovered groups from all hyperparameter settings, bin them by group size, and compute (left) the average FNMR@FMR=$10^{-3}$ of the top-100 most biased groups (higher = more bias revealed), and (right) the average intra-group semantic distance of the top-100 most coherent groups (lower = more semantically consistent). LFA consistently identifies groups with both higher bias exposure and stronger semantic coherence across all evaluated model architectures.

Error rates are computed from the impostor and genuine comparisons: EER (Equal Error Rate), FNMR (False Non-Match Rate), and FMR (False Match Rate). The FNMR comes from the genuine distribution, which is similar across groups, but the impostor distribution varies greatly, so we focus on the FMR. Figure 3c shows FMR curves for the different groups evaluating the ArcFace model. The FMR curves show a clear disparity across groups, which is the sign of bias in biometric systems. The rest of the models (CosFace, ElasticFace, and PartialFC) have a similar score distributions and thus similar error rate curves (details are provided in Appendix B).

Table 2 reports false match rates (FMR) at a fixed threshold ($t = 0.2$) for the LFA-discovered groups (A–D), compared against groups defined using explicit attribute annotations. Across all four backbones, LFA groups exhibit markedly higher FMR than a random subset, in some cases exceeding the baseline by more than $50\times$. Interestingly, although the exact groups of LFA groups do not fully align with those formed via attribute annotations, due to limitations in VLM attribute coverage (e.g., no attribute for black-and-white photos), the trends remain consistent: our method tends to uncover even more homogeneous groups, always resulting in higher FMR. (see Appendix B for full bias evaluation metrics and confidence intervals.)

We performed hyperparameter ablation experiments for all baseline algorithms and *LFA* to facilitate a meaningful comparison. The results include the average of top-10 FNMR groups, with confidence intervals, and indicate similar performance trends. This comparison allows us to assess the relative strengths and weaknesses of each method, highlighting how hyperparameter adjustments impact the discovery of high FNMR groups. (Appendix Figures 11–16.)

### 4.5 COMPARISON WITH UNSUPERVISED BASELINES

The objective of *LFA* is to identify subpopulations that are simultaneously highly biased (i.e., exhibit large performance disparities) and semantically coherent (i.e., consist of visually or attribute-wise similar samples). To contextualize *LFA*'s effectiveness, we compare it against a set of unsupervised baselines: *Nearest Neighbors*, *k-means*, *spherical k-means*, and *DBSCAN*, using two metrics: (a) FNMR@FMR= $10^{-3}$ (higher error indicates stronger bias), and (b) average intra-group semantic distance (lower indicates higher semantic coherence).

To ensure comparability across methods with very different clustering behaviors, we adopt the following evaluation protocol. For each algorithm, we run all combinations of hyperparameters, collect all discovered groups, and bin groups by their size. Within each size bin, we rank groups by FNMR and semantic coherence, and report the top-100 groups in each size. This controls for the fact that clustering algorithms can produce very large or very small clusters, which can trivially improve or worsen these metrics.

Figure 4 summarizes these results across all four face-recognition models. LFA consistently discovers (i) the groups with the highest FNMR, indicating that it surfaces subpopulations that are more strongly biased than those found by any baseline, and (ii) the most semantically coherent groups, reflected in lower intra-group attribute distance. Importantly, these trends hold across all evaluated group sizes and model backbones.

## 5 CONCLUSION AND DISCUSSION

In this work, we introduced Latent Feature Alignment (*LFA*), a simple yet effective method for discovering semantically coherent and interpretable subpopulations directly from face recognition embeddings. Unlike traditional clustering methods that rely purely on local distance metrics, *LFA* exploits latent directions in the embedding space, leading to groups that are both more coherent and more interpretable. Our experimental evaluation across CelebA and RFW demonstrates three main contributions.

First, we showed that *LFA* consistently produces groups with higher semantic coherence than $k$-means, spherical $k$-means, DBSCAN and nearest-neighbor search, across four state-of-the-art recognition models (ArcFace, CosFace, ElasticFace, PartialFC) and two widely used benchmarks (RFW, CelebA) (Section 4.2 and Appendix E). Second, we demonstrated that the dominant directions discovered by *LFA* correspond to meaningful semantic attributes, as evidenced by both qualitative traversals and quantitative attribute probability trends using arc2face and InfiniteYou (Section 4.3 and Appendix D). These results highlight the interpretability of the latent space, even when derived from discriminative embeddings not trained for generative editing (such as GANs or autoencoders). Third, we validated that the discovered groups present systematic disparities in recognition performance. Specifically, subpopulations identified by *LFA* exhibit higher error rates consistent with trends observed using explicit attribute annotations (Section 4.4 and Appendix B).

Our findings open several avenues for future work. While *LFA* uncovers meaningful latent directions, we also observed non-linear effects or entanglement between attributes, suggesting that more refined methods could improve disentanglement. Moreover, although we validated robustness with VLM-assisted annotations, the absence of certain attributes (e.g., black-and-white images) limits the completeness of comparisons. Extending the annotation coverage and applying *LFA* to larger and more diverse datasets could provide a fuller picture of its potential.

In summary, *LFA* offers a practical and interpretable tool for analyzing embedding spaces, providing insights into both semantic structure and hidden biases. By broadening the scope of fairness auditing to domains where labeled attributes are limited, costly, or incomplete, LFA facilitates new methodologies for bias detection in high-stakes applications. We hope this work contributes to the growing literature on transparent and accountable machine learning.

## 6 LIMITATIONS

**Initialization.** The quality of *LFA* groups depends on initialization. Future work could investigate exhaustive exploration of initial combinations can yield more precise groups (Appendix D).

**Observational Nature and Confounders.** Our method is observational, relying solely on naturally occurring data. As such, it cannot disentangle semantic attributes from nuisance factors such as illumination, pose, or resolution. This limitation is inherent to any bias analysis performed without controlled interventions.

**Linear Approximation of the Latent Space.** *LFA* assumes locally linear directions in a highly dimensional embedding space. This linearization may overlook non-linear or entangled trajectories in the manifold, leading to partial or imperfect attribute representation.

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

# A   RFW ANNOTATION AND VALIDATION

Table 3: OpenCompass VLM evaluation results of our selected Visual Language Models (VLMs) for annotating RFW. * Denotes the score of the unquantized version.

| Method | Params (B) | Language Model | Vision Model | Avg. Score |
|---|---|---|---|---|
| InternVL3-38B-AWQ | 38.4 | Qwen2.5-32B | InternViT-6B-v2.5 | 77.8* |
| Ovis2-34B-GPTQ-Int8 | 34.9 | Qwen2.5-32B | AIMv2-1B | 76.5* |
| SAIL-VL-1.6-8B | 8.33 | Qwen2.5-7B | AIMv2 Huge | 73.6 |
| Ola-7b | 8.88 | Qwen2.5-7B | Oryx-ViT | 72.6 |
| Qwen2.5-VL-32B-AWQ | 32.5 | Qwen2.5-32B | QwenViT | - |

Given the noise of CelebA attribute annotations, we decided to provide more robust evidence of the better clustering. For that we annotated RFW with five state of the art VLMs: InternVL3-38B-AWQ (Chen et al., 2024), Ola-7b (Liu et al., 2025), Ovis2-34B-GPTQ-Int8 (Lu et al., 2024), Qwen2.5-VL-32B-Instruct-AWQ (Bai et al., 2025), and SAIL-VL-1d6-8B (Dong et al., 2025).

The models were chosen using the benchmark on the OpenCompass VLM leaderboard[1]. The top 20 open source models belong to the 5 VLM families, and they vary in size. We chose one model from each family, 3 being of size ∼32B, and 2 of size 7B. The 32B models are quantized versions of the models (to fit a GPU). Table 3 shows the models' vision and language backbones, number of parameters, and average score. The Qwen2.5-VL-32B model does not appear in the leaderboard.

The following prompt was used to elicit facial attribute classification from the VLMs:

```
You are an image analysis tool specialized in facial attribute classification.
For the provided face image, output a JSON object with the following attributes:

{
  "gender": ["male", "female"],
  "age": ["young", "middle-aged", "senior"],
  "skin_color": ["light", "medium", "dark"],
  "ancestry": ["asian", "south_asian", "black", "latino/hispanic", "middle_eastern",
               "white", "indigenous"],
  "hair_color": ["black", "brown", "red", "blonde", "gray", "other"],
  "bangs": ["yes", "no"],
  "bald": ["yes", "no"],
  "beard": ["no", "mustache", "stubble", "full"],
  "glasses": ["no", "regular", "sun"],
  "headwear": ["no", "beanie", "cap", "hat", "headband", "hijab", "helmet", "turban"]
}

Ensure the labeling is based on visible evidence only. If an attribute is unclear,
return "unknown".

Only output the JSON without any additional explanation or text.
Example JSON output:
{
  "gender": "female",
  "age": "middle-aged",
  "skin_color": "light",
  "ancestry": "asian",
  "hair_color": "black",
  "bangs": "no",
  "bald": "no",
  "beard": "no",
  "glasses": "sun",
  "headwear": "beanie"
}
```

Table 4 breaks down each attribute annotation showing how many samples fall into each class, both as a raw count and as a percentage of the total dataset. There is also an "Unknown" category for each attribute, which indicates how many images had no consensus among the models for that particular attribute. We quantify labeling bias computing compute agreement scores. These scores reflect the proportion of VLMs that agreed on the final label. For example, for the final label "mustache" in the 'beard' attribute of an image, if 3 VLMs say 'mustache', 1 says 'stubble' and 1 says 'no', the

[1]https://huggingface.co/spaces/opencompass/open_vlm_leaderboard

agreement ratio for that image and that attribute is 3/5 = 0.6. Table 4 presents mean agreement and standard deviation by attribute. Most classes show a high degree of agreement. Only *Hair Color* and *Ancestry* show less agreement for less frequent classes like *Latino/Hispanic*, *Indigenous*, or *Other*.

Table 4: Attribute annotations and agreement scores (mean and standard deviation) of the RFW dataset using five state-of-the-art Visual Language Models. *Unknown* class means there was no consensus among models. Total samples: 40,607.

| Category | Class | Count | % | Mean Agr. | Std Agr. |
|---|---|---|---|---|---|
| Age | Middle-aged | 19,072 | 47.0 | 0.88 | 0.15 |
| | Young | 16,144 | 39.8 | 0.90 | 0.15 |
| | Senior | 5,312 | 13.1 | 0.92 | 0.14 |
| | *Unknown* | 79 | 0.2 | | |
| Ancestry | Black | 10,396 | 25.6 | 0.99 | 0.04 |
| | White | 10,161 | 25.0 | 0.95 | 0.11 |
| | Asian | 9,547 | 23.5 | 0.98 | 0.07 |
| | South Asian | 9,191 | 22.6 | 0.93 | 0.12 |
| | Latino/Hispanic | 379 | 0.9 | 0.67 | 0.10 |
| | Middle Eastern | 146 | 0.4 | 0.70 | 0.12 |
| | Indigenous | 18 | 0.0 | 0.81 | 0.18 |
| | *Unknown* | 769 | 1.9 | | |
| Bald | No | 37,553 | 92.5 | 0.98 | 0.07 |
| | Yes | 3,047 | 7.5 | 0.84 | 0.15 |
| | *Unknown* | 7 | 0.0 | | |
| Bangs | No | 35,280 | 86.9 | 0.94 | 0.11 |
| | Yes | 5,297 | 13.0 | 0.86 | 0.16 |
| | *Unknown* | 30 | 0.1 | | |
| Beard | No | 30,356 | 74.8 | 0.91 | 0.12 |
| | Stubble | 3,940 | 9.7 | 0.74 | 0.15 |
| | Mustache | 3,677 | 9.1 | 0.89 | 0.16 |
| | Full | 1,263 | 3.1 | 0.77 | 0.16 |
| | *Unknown* | 1371 | 3.4 | | |
| Gender | Male | 30,191 | 74.3 | 1.00 | 0.04 |
| | Female | 10,400 | 25.6 | 0.99 | 0.06 |
| | *Unknown* | 16 | 0.0 | | |
| Glasses | No | 34,589 | 85.2 | 1.00 | 0.03 |
| | Regular | 4,914 | 12.1 | 0.98 | 0.08 |
| | Sun | 1,080 | 2.7 | 0.94 | 0.12 |
| | *Unknown* | 24 | 0.1 | | |
| Hair Color | Black | 26,114 | 64.3 | 0.96 | 0.11 |
| | Brown | 5,795 | 14.3 | 0.82 | 0.15 |
| | Gray | 5,529 | 13.6 | 0.91 | 0.14 |
| | Blonde | 1,780 | 4.4 | 0.92 | 0.14 |
| | Red | 935 | 0.8 | 0.84 | 0.16 |
| | Other | 129 | 0.3 | 0.65 | 0.06 |
| | *Unknown* | 942 | 2.3 | | |
| Headwear | No | 35,543 | 87.5 | 0.99 | 0.04 |
| | Cap | 1,991 | 4.9 | 0.92 | 0.14 |
| | Hat | 1,089 | 2.7 | 0.87 | 0.17 |
| | Headband | 513 | 1.3 | 0.81 | 0.16 |
| | Beanie | 304 | 0.7 | 0.89 | 0.16 |
| | Turban | 262 | 0.6 | 0.94 | 0.13 |
| | Helmet | 255 | 0.6 | 0.89 | 0.14 |
| | Hijab | 199 | 0.5 | 0.91 | 0.15 |
| | *Unknown* | 451 | 1.1 | | |
| Skin Tone | Medium | 15,499 | 38.2 | 0.76 | 0.12 |
| | Light | 13,701 | 33.7 | 0.91 | 0.15 |
| | Dark | 10,952 | 27.0 | 0.97 | 0.09 |
| | *Unknown* | 455 | 1.1 | | |

## ANNOTATION MERGING VIA MAJORITY VOTING

To merge the predictions of the five VLMs into a single robust annotation per attribute, we used a majority voting scheme that explicitly ignores unknown values. For each image and attribute, we collected the predictions from all five models and discarded any labeled as unknown. The final consensus label was defined as the label that received strictly more than half of the valid (i.e., non-unknown) predictions. If no such label existed—either due to a tie or insufficient agreement—the annotation was marked as unknown.

Table 5: **Validation of automatic attribute annotations against a human-annotated 1K sample.**
Each row reports the Match Rate (fraction of exact matches) and Cohen's $\kappa$ (agreement corrected for chance). High $\kappa$ values indicate substantial-to-almost-perfect agreement for most attributes.

| Attribute | Match Rate ($\uparrow$) | Cohen's $\kappa$ ($\uparrow$) |
|---|---|---|
| Age | 0.8840 | 0.8085 |
| Ancestry | 0.9473 | 0.9306 |
| Bald | 0.9750 | 0.8312 |
| Bangs | 0.9560 | 0.7961 |
| Beard | 0.8918 | 0.7462 |
| Gender | 0.9900 | 0.9741 |
| Glasses | 0.9920 | 0.9694 |
| Hair Color | 0.9007 | 0.8183 |
| Headwear | 0.9668 | 0.8630 |
| Skin Color | 0.9304 | 0.8941 |

Formally, let $A_i^j$ be the label predicted by model $i$ for attribute $j$, and define $V = \{A_i^j \mid A_i^j \neq$ unknown$\}$ as the set of valid predictions for that attribute. Let $n = |V|$ denote the number of valid votes. The consensus label $C^j$ is then given by:

$$C^j = \begin{cases} \ell \in V, & \text{if } \mathrm{count}(\ell) > \frac{n}{2} \\ \texttt{unknown}, & \text{otherwise} \end{cases}$$

This procedure ensures that only strong and unambiguous agreement among the models leads to an assigned label, while also allowing for occasional uncertainty (expressed via unknown) improving the reliability of the annotations used in our clustering analysis. We emphasize that the labels obtained via majority voting among VLMs serve as weak annotations and are not ground truth. In rare cases (fewer than 0.1% of samples), images with multiple faces were manually disambiguated by selecting the most centrally located face.

HUMAN ANNOTATION VALIDATION

We validated the automatic attribute annotations against a human-annotated sample. Table 5 shows per-attribute raw match rate (the fraction of exact label matches) and Cohen's $\kappa$ (agreement corrected for chance). The validation sample contains 1,000 matched images per attribute. For binary attributes with strong class balance (e.g., gender, glasses), raw accuracy and $\kappa$ are both very high (*gender*: 99% match, $\kappa = 0.974$; *glasses*: 99% match, $\kappa = 0.969$), indicating near-perfect agreement. Multi-class attributes (*age*, *skin color*, *ancestry*, *hair color*) also show very strong agreement (e.g., *ancestry* $\kappa = 0.931$), indicating the automatic labels are reliable for downstream analyses.

Cohen's $\kappa$ measures inter-rater reliability for qualitative (categorical) classes and controls for chance agreement and class imbalance; it should be preferred to raw match rate when classes are uneven. Per-class metrics (precision/recall/F1) and confusion matrices (provided in the supplement) reveal that some low-support classes (for example, the *Latino/Hispanic* ancestry class and several rare *Hair-color* categories) have poor per-class precision/recall despite high overall accuracy. These small-class failures do not substantially affect the aggregate statistics but are important to note when analyzing model behaviour for specific subgroups.

The low agreement for the attribute *Beard* is due to the fact that the human annotators considered goatees and other beard styles fall into the *Stubble* category, while VLM did not. Figure 5 include the full confusion matrices of subcategory-level matching.

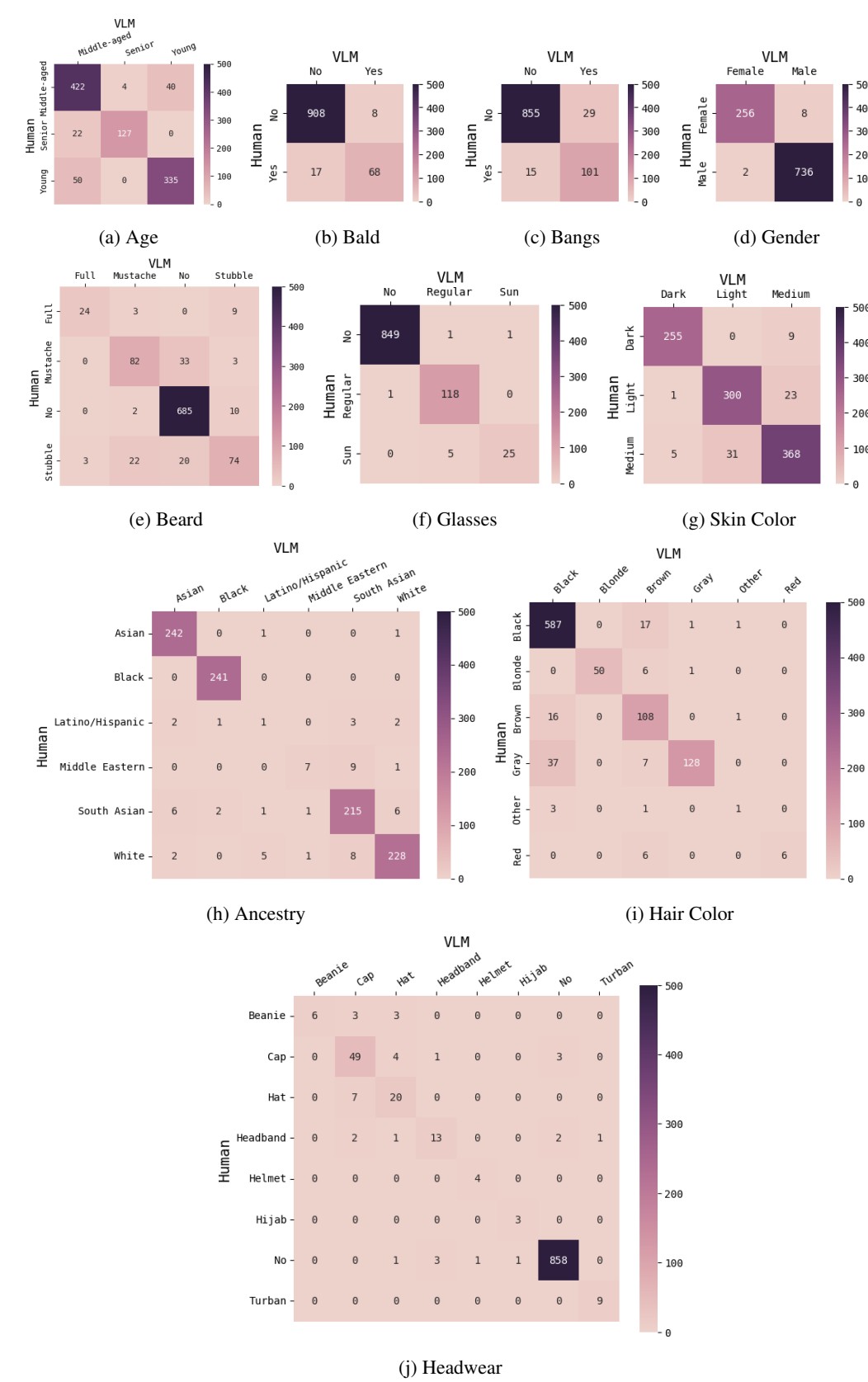

Figure 5: Confusion matrices for each attribute between human annotations and VLM annotations.

# B BIAS EVALUATION

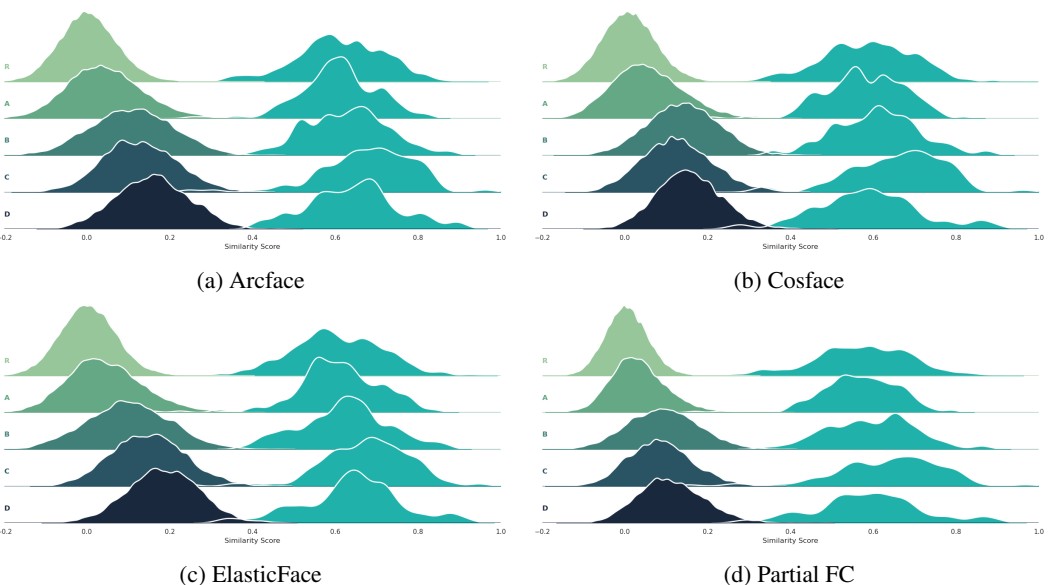

(a) Arcface

(b) Cosface

(c) ElasticFace

(d) Partial FC

Figure 6: Distribution of cosine similarity scores for impostor and genuine comparisons within the same group.

Table 6: Performance (EER and FNMR) of the face recognition models across the discovered groups (A-D), with standard deviation across groups (lower means fairer, less biased). Each group includes the number of images and comparisons.

| | EER | | | | |
|---|---|---|---|---|---|
| **Model** | **Group A** (130 / 8k) | **Group B** (143 / 10k) | **Group C** (96 / 4k) | **Group D** (100 / 5k) | $\sigma$ |
| ArcFace | $8.70 \times 10^{-3}$ | $4.00 \times 10^{-4}$ | $2.42 \times 10^{-2}$ | $1.00 \times 10^{-3}$ | 0.0096 |
| CosFace | $5.10 \times 10^{-3}$ | $6.40 \times 10^{-3}$ | $1.83 \times 10^{-2}$ | $1.57 \times 10^{-2}$ | 0.0057 |
| ElasticFace | $8.20 \times 10^{-3}$ | $1.40 \times 10^{-3}$ | $1.22 \times 10^{-2}$ | $1.89 \times 10^{-2}$ | 0.0063 |
| Partial FC | $6.30 \times 10^{-3}$ | $1.20 \times 10^{-3}$ | $1.94 \times 10^{-2}$ | $1.78 \times 10^{-2}$ | 0.0077 |
| | **FNMR @ FMR=1%** | | | | |
| **Model** | **Group A** (130 / 8k) | **Group B** (143 / 10k) | **Group C** (96 / 4k) | **Group D** (100 / 5k) | $\sigma$ |
| ArcFace | $1.15 \times 10^{-2}$ | 0.00 | $2.08 \times 10^{-2}$ | 0.00 | 0.0087 |
| CosFace | $5.70 \times 10^{-3}$ | 0.00 | $3.12 \times 10^{-2}$ | $2.52 \times 10^{-2}$ | 0.0130 |
| ElasticFace | $1.15 \times 10^{-2}$ | 0.00 | $2.08 \times 10^{-2}$ | $3.36 \times 10^{-2}$ | 0.0123 |
| Partial FC | $5.70 \times 10^{-3}$ | 0.00 | $2.08 \times 10^{-2}$ | $1.68 \times 10^{-2}$ | 0.0083 |
| | **FNMR @ FMR=0.1%** | | | | |
| **Model** | **Group A** (130 / 8k) | **Group B** (143 / 10k) | **Group C** (96 / 4k) | **Group D** (100 / 5k) | $\sigma$ |
| ArcFace | $2.30 \times 10^{-2}$ | 0.00 | $3.12 \times 10^{-2}$ | $4.20 \times 10^{-2}$ | 0.0154 |
| CosFace | $1.72 \times 10^{-2}$ | $1.97 \times 10^{-2}$ | $4.17 \times 10^{-2}$ | $7.56 \times 10^{-2}$ | 0.0234 |
| ElasticFace | $2.30 \times 10^{-2}$ | $6.60 \times 10^{-3}$ | $3.12 \times 10^{-2}$ | $8.40 \times 10^{-2}$ | 0.0290 |
| Partial FC | $1.72 \times 10^{-2}$ | $6.60 \times 10^{-3}$ | $3.12 \times 10^{-2}$ | $8.40 \times 10^{-2}$ | 0.0297 |

We evaluate face recognition performance across some discovered groups (A-D) by forming genuine and imposter pairs exclusively within each group. Genuine pairs are pairs of images belonging to the

Table 7: False Match Rates (FMR) with 95% confidence intervals at a fixed threshold ($t = 0.2$) for each demographic group across four face recognition methods. Confidence intervals are computed via bootstrap resampling (1,000 iterations) at the image level.

| Method | Group R | Group A | Group B | Group C | Group D |
|---|---|---|---|---|---|
| ArcFace | 0.0057 ± 0.002 | 0.0522 ± 0.013 | 0.1798 ± 0.021 | 0.2167 ± 0.036 | 0.3181 ± 0.043 |
| CosFace | 0.0045 ± 0.002 | 0.0521 ± 0.014 | 0.2447 ± 0.027 | 0.1952 ± 0.038 | 0.2780 ± 0.036 |
| ElasticFace | 0.0043 ± 0.002 | 0.0569 ± 0.015 | 0.1676 ± 0.021 | 0.2861 ± 0.044 | 0.4365 ± 0.050 |
| PartialFC | 0.0016 ± 0.001 | 0.0161 ± 0.007 | 0.1060 ± 0.016 | 0.0809 ± 0.022 | 0.1400 ± 0.033 |

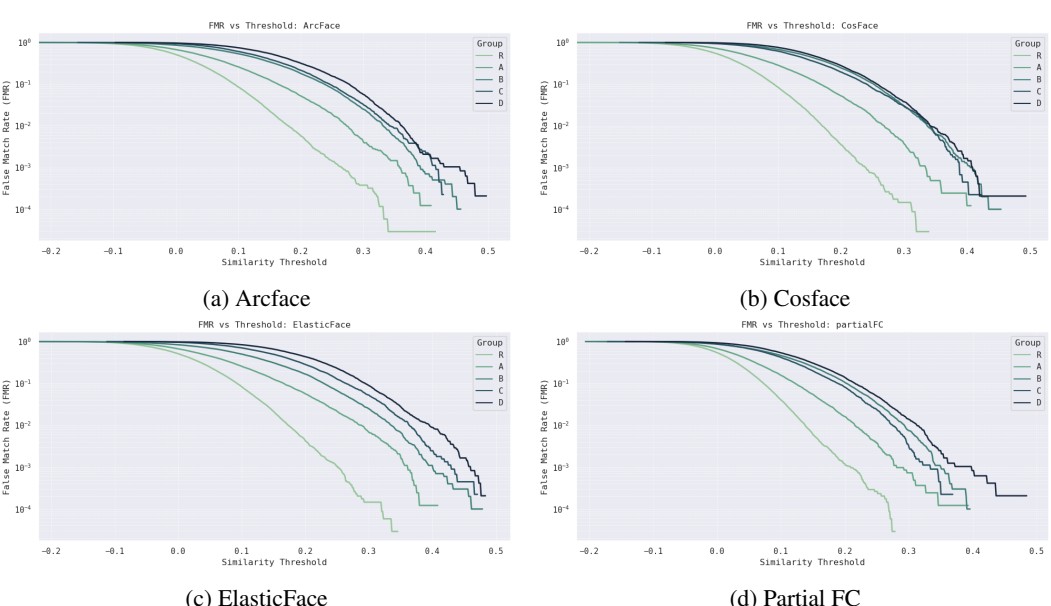

(a) Arcface

(b) Cosface

(c) ElasticFace

(d) Partial FC

Figure 7: False Match Rate (FMR) curves for each face recognition model for all groups.

same person and imposter pairs are formed by pairing images from different identities. Similarity scores are computed using the cosine similarity of the corresponding face embeddings.

Figure 6 shows the distribution of cosine similarity scores per model for genuine and imposter pairs within each group. The right distributions (gray) corresponds to genuine pairs (higher similarity values), while the left distribution (colored) corresponds to impostor pairs (lower similarity values). Darker colors indicate groups with higher error (imposter distributions shifted towards genuine), while lighter colors indicate groups with lower error.

From the resulting score distributions, we compute:

**Equal Error Rate (EER)**: the operating point where false match rate (FMR) equals false non-match rate (FNMR).

**FNMR at FMR=1% and 0.1%**: the FNMR (False Non-Match rate) value when the decision threshold is set to yield an FMR (False Match Rate) of 1% and 0.1%.

Table 6 summarizes the verification metrics for all groups. Sample sizes vary per group and are reported alongside EER, FNMR@1%FMR, and FNMR@0.1%FMR. The standard deviation ($\sigma$) between A-D groups is a common intuitive proxy for bias in biometric systems as it indicates the statistical measure of variance across groups — higher $\sigma$ means more disparity. FNMR@0.1%FMR means the false non-match rate at a fixed threshold for which the false match rate is 0.1%.

Although we observe that Groups C and D exhibit higher EER and FNMR than A and B, confirming the presence of bias in model performance for these groups, these groups lack the necessary number of comparisons for detailed measurements (notice the zeroes results for group B in FNMR @ FMR=1%).

Although Groups C and D exhibit higher EER and FNMR compared to Groups A and B, indicating bias in the model's performance, they groups lack sufficient comparative data for detailed analysis at high error rates. For example, Group B has FNMR values of zero at FMR = 1%. To address this limitation, Table 7 reports false match rates (FMR) at a fixed threshold ($t = 0.2$) with confidence intervals to assess statistical significance. Confidence intervals are computed via bootstrap resampling at the image level (1,000 iterations).

When evaluating face recognition systems at a fixed False Match Rate (FMR) of 0.1%, the decision threshold is chosen to allow only 1 in 1,000 impostor comparisons to be incorrectly accepted. The corresponding False Non-Match Rate (FNMR) is then computed over genuine comparisons using this threshold. However, when the number of genuine pairs is small—for example, approximately 120, like in our case—each false non-match has a disproportionately large effect on the FNMR estimate. Specifically, each false rejection increases the FNMR by roughly 0.83% (i.e., 1/120). This implies that the FNMR can only take on discrete values, such as 0%, 0.83%, 1.67%, etc., making it difficult to make fine-grained measurements. As a result, observed differences in FNMR at low FMRs may not be statistically significant or practically meaningful. Small improvements reported at this operating point could simply be due to random variation rather than true differences in system performance.

A more robust analysis of imposter distributions (in the order of thousands of scores) is to compute the FMR (False Match Rate) curve group-wise as a function of the threshold. Figure 7 shows the FMR curves for each face recognition model—ArcFace, CosFace, ElasticFace, and Partial FC—disaggregated by group. These curves illustrate how the False Match Rate varies across groups with the decision threshold, highlighting disparities between different groups.

## C  TRAVERSALS

To reconstruct images from facial embeddings, we use arc2face (Papantoniou et al., 2024), which synthesizes high-quality, identity-preserving images given ArcFace (Deng et al., 2019) embeddings.

Since ArcFace embeddings lie on the unit hypersphere, we cannot apply edits via direct vector addition. Instead, we use spherical linear interpolation (slerp) (Shoemake, 1985) to interpolate between embeddings. This approach respects the geometry of the hypersphere and provides fine-grained control over the edit intensity.

In contrast to linear interpolation or simple vector averaging—which may produce off-manifold results or require manual tuning of edit strength depending on the embedding and direction—slerp offers a principled mechanism for traversal along geodesics.

The slerp between two unit vectors $p_0$ and $p_1$ is defined as:

$$\text{Slerp}(p_0, p_1; t) = \frac{\sin((1-t)\theta)}{\sin\theta} p_0 + \frac{\sin(t\theta)}{\sin\theta} p_1$$

where $t \in [0, 1]$ is the interpolation parameter and $\theta = \cos^{-1}(p_0 \cdot p_1)$ is the angle between $p_0$ and $p_1$. The interpolation smoothly transitions from $p_0$ to $p_1$ along the great arc on the hypersphere.

For the results shown in Figure 1a, we used a fixed interpolation strength of $t = -0.5$ to subtract and visualize the edit direction. For Figure 1b, the interpolation strength had to be adjusted between $t = 0.45$ and $t = 0.5$ depending on the embedding and direction. In a few rare cases, a value as low as $t = 0.4$ was required to ensure plausible and visually coherent outputs.

### DECODER-AGNOSTIC TRAVERSALS

To ensure that the interpretability of LFA-discovered directions is not specific to Arc2Face, we repeated the traversal experiments using InfiniteYou (InfU) (Jiang et al., 2025), a recently proposed embedding-conditioned synthesis model. InfU differs substantially from Arc2Face in both its encoder and decoder components:

**Encoder**: IR-SE50 (instead of ArcFace-R100 trained on Glint360K).

**Decoder**: Flux diffusion model (instead of Stable Diffusion).

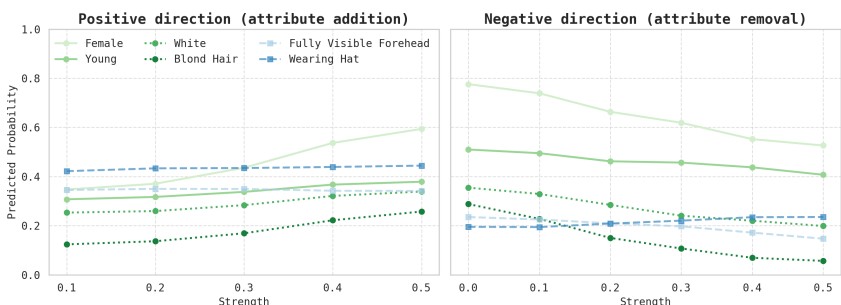

Figure 8: **Attribute probability vs traversal strength** using InfiniteYou (Jiang et al., 2025). Predicted probabilities from an attribute classifier for images traversed along the latent direction of Group A (young, white, blonde, female). Relevant attributes (e.g., *Blond Hair, Young, Female, White*) change monotonically with traversal strength, while unrelated attributes (e.g., *Wearing Hat, Fully Visible Forehead*) remain stable.

We selected the latent direction corresponding to Group A (young, white, blonde, female) and performed traversals in both the positive (attribute addition) and negative (attribute removal) directions. Following Section 4.3, we measured facial-attribute probabilities across interpolation strengths using an attribute classifier trained on CelebA following Lingenfelter & Hand (2021).

Figure 8 shows qualitative reconstructions along the latent direction. Quantitatively, InfU replicates the same patterns observed with Arc2Face: traversals yield consistent and monotonic changes in the key attributes (*Blond Hair, Young, Female, White*), while unrelated attributes, such as *Wearing Hat* and *Fully Visible Forehead*, remain stable.

The small deviations observed in the negative traversal reinforce an important point discussed in the main paper: embedding-space attribute directions reflect interactions between the encoder and decoder, and certain regions of the encoder output space may be poorly modeled by the decoder. Given that InfU uses a different encoder and a different diffusion architecture, the close agreement in direction semantics across the two systems provides strong evidence that LFA discovers genuine semantic directions intrinsic to the embedding space, rather than artifacts tied to a specific generator.

Interestingly, the negative traversal shows a slightly different behavior: *Young* decreases mildly. This deviation is informative: it suggests that the decoder plays a substantial role in how attribute changes manifest. The encoder's output space is already complex, and once it becomes the input space of the decoder, additional modeling constraints and mismatches can arise. As a result, certain regions of the embedding space may not be equally well captured by the decoder, producing more asymmetries such as the one observed here. Still, despite using a different encoder and an entirely different diffusion architecture, InfU yields overall direction semantics that closely match those obtained with Arc2Face.

## D  ABLATION STUDY – QUALITATIVE RESULTS

This section presents illustrative examples of high-quality groups formed using *LFA* on the CelebA dataset. All groups were constructed using embeddings extracted with Partial FC, and a threshold $\tau = 0.35$ for *LFA* algorithm. The examples highlight attribute-based groups that exhibit strong semantic coherence, in part because all initial images belong to different identities.

Figures 9–19 display groups representing the following categories: males with turbans, males with baseball helmets, dark-skinned bald males, and light-skinned bald males. In each figure, the green box in the top-left corner indicates the initial group. Images are displayed in left-to-right, top-to-bottom sequence, reflecting the order in which they were added to the group.

We focus on CelebA due to its high number of images per identity, which increases the likelihood that grouping will be influenced by identity-related features, and makes it more complicated. When an initial group consists of images from the same identity, the latent direction in the embedding

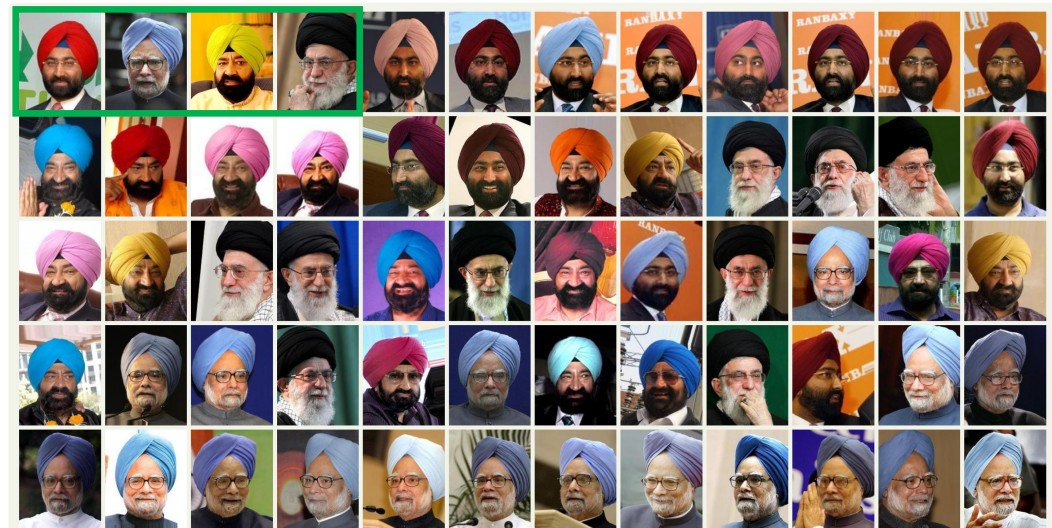

Figure 9: CelebA images of males with turban. Initial group in green.

space may not yet be well defined. In such cases, expansion tends to favor look-alike identities rather than consistent attribute alignment.

To illustrate this, Figure 10 shows a group formed from four images of Asian females wearing tennis caps, all from different identities. The resulting group consistently includes images of individuals wearing (mostly white) tennis caps, indicating that the latent direction was successfully aligned with the intended visual attribute.

By contrast, Figures 20–22 show examples where the initial group consisted of a single image. In these cases, the grouping is driven primarily by identity similarity. While this is not inherently undesirable, it leads to groups where the dominant shared feature may be an identity-specific facial trait.

These examples validate the method's ability to discover visually meaningful groups when initialized with diverse identity faces. They also support our claim that exhaustive or informed initial selection improves grouping quality.

We observed some groups that reflect non-demographic semantics (e.g., a "black-and-white photograph" group combining demographic and stylistic factors, shown in Figs. 1a and 1b). However, we did not observe groups dominated by pose, or blur. Groups consistently reflect human-interpretable factors such as age, ancestry, hair/attire, colorization style, and domain.

# E ABLATION STUDY – QUANTITATIVE RESULTS

We evaluate the effectiveness of *LFA* versus *k-means* clustering on the CelebA dataset using four face recognition models: ArcFace, CosFace, ElasticFace, and PartialFC.

A fair comparison between *LFA* and *k-means* requires that both methods produce groups of similar size (i.e., similar average number of images per group). This is crucial because larger groups tend to exhibit greater variation, which increases the average intra-group attribute distance.

Therefore, instead of selecting a fixed value of $k$ or threshold arbitrarily, we adjust $k$ (for *k-means*) and $\tau$ (for *LFA*) to yield clusters with comparable average sizes. For example, a setting with $k = 2000$ typically yields clusters with approximately 100 images. We adopt the same principle here and align group sizes across methods to allow meaningful comparison.

In Table 8, we report the average intra-group attribute distance (lower is better) for each method, at three different cluster size settings (50, 100, and 200 images per group). For each setting, we bold

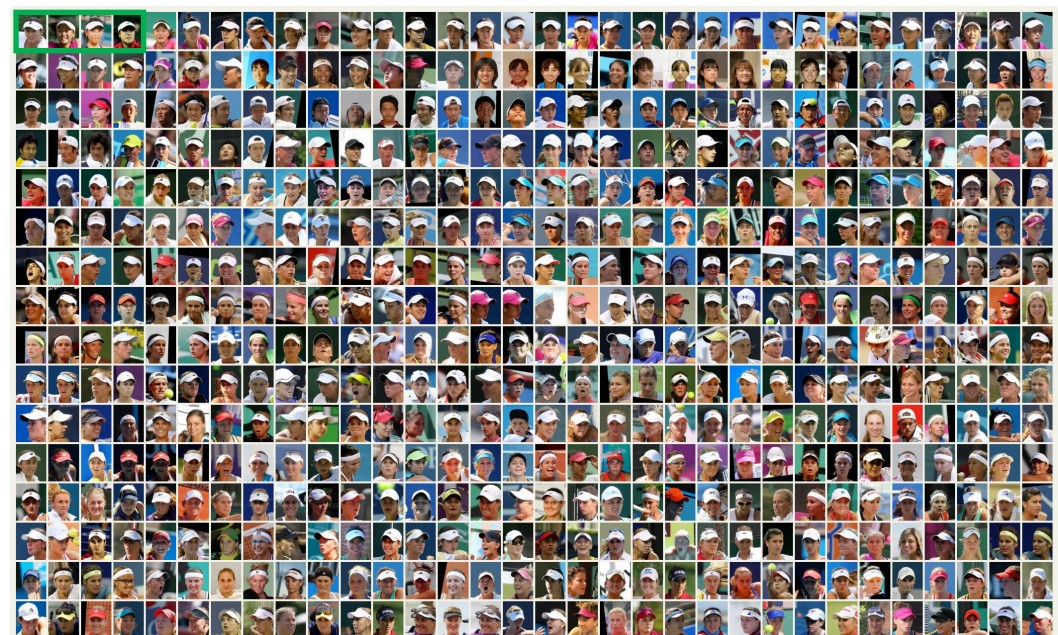

Figure 10: CelebA images of asian females with a tennis cap. Initial group in green.

Table 8: Comparison of intra-group attribute distances between *k-means* and *LFA* at different levels of $N$ (average number of samples) in the CelebA database. Thresholds $\tau$ for *LFA* and $k$ for *k-means* are shown in parentheses. Bold values indicate better performance between the two methods under comparable group sizes.

| Method | N (avg) | ArcFace ($\tau$ \| k) | CosFace ($\tau$ \| k) | ElasticFace ($\tau$ \| k) | PartialFC ($\tau$ \| k) |
|--------|---------|----------------------|----------------------|---------------------------|-------------------------|
| K-Means | ~50 | 6.58 (4000) | 6.55 (4000) | **6.04** (4000) | **6.48** (4000) |
| LFA | ~50 | **6.09** (0.42) | **6.08** (0.42) | 6.21 (0.35) | 6.13 (0.40) |
| K-Means | ~100 | 7.65 (2000) | 7.61 (2000) | 7.55 (2000) | 7.50 (2000) |
| LFA | ~100 | **6.42** (0.40) | **6.41** (0.40) | **6.57** (0.33) | **6.28** (0.39) |
| K-Means | ~200 | 8.52 (1000) | 8.49 (1000) | 8.38 (1000) | 8.28 (1000) |
| LFA | ~200 | **6.87** (0.38) | **6.95** (0.38) | **7.03** (0.31) | **6.64** (0.37) |

the lower (better) value between *LFA* and *k-means* for each model. Thresholds for *LFA* and values of $k$ for *k-means* are shown in parentheses.

Figures 11 and 12 report an ablation study across cosine-graph thresholds for pre-clustering initializations and projection thresholds $\tau$ and compares to random seeds (*without* similarity graph pre-clustering). The algorithm is applied to RFW embeddings obtained using the four face recognition models. These plots contain, average of top-10 FNMR groups, average of top-100 semantic coherence (attribute distance) groups, average size of discovered groups, and total number of groups discovered. FNMR is computed at the FMR = $10^{-3}$. We control for group size ($\geq 20$ identities) to ensure statistical validity. The top-10 and top-100 allow to see the groups that matter.

Figures 13 - 15 show the results for Nearest Neighbors, k-means, spherical k-means, and DBSCAN, with different parameter values, applied to RFW embeddings obtained using the four face recognition models.

Across all models, LFA discovers groups with (i) the highest FNMR and (ii) the best semantic coherence both among all methods, relative to group/cluster size.

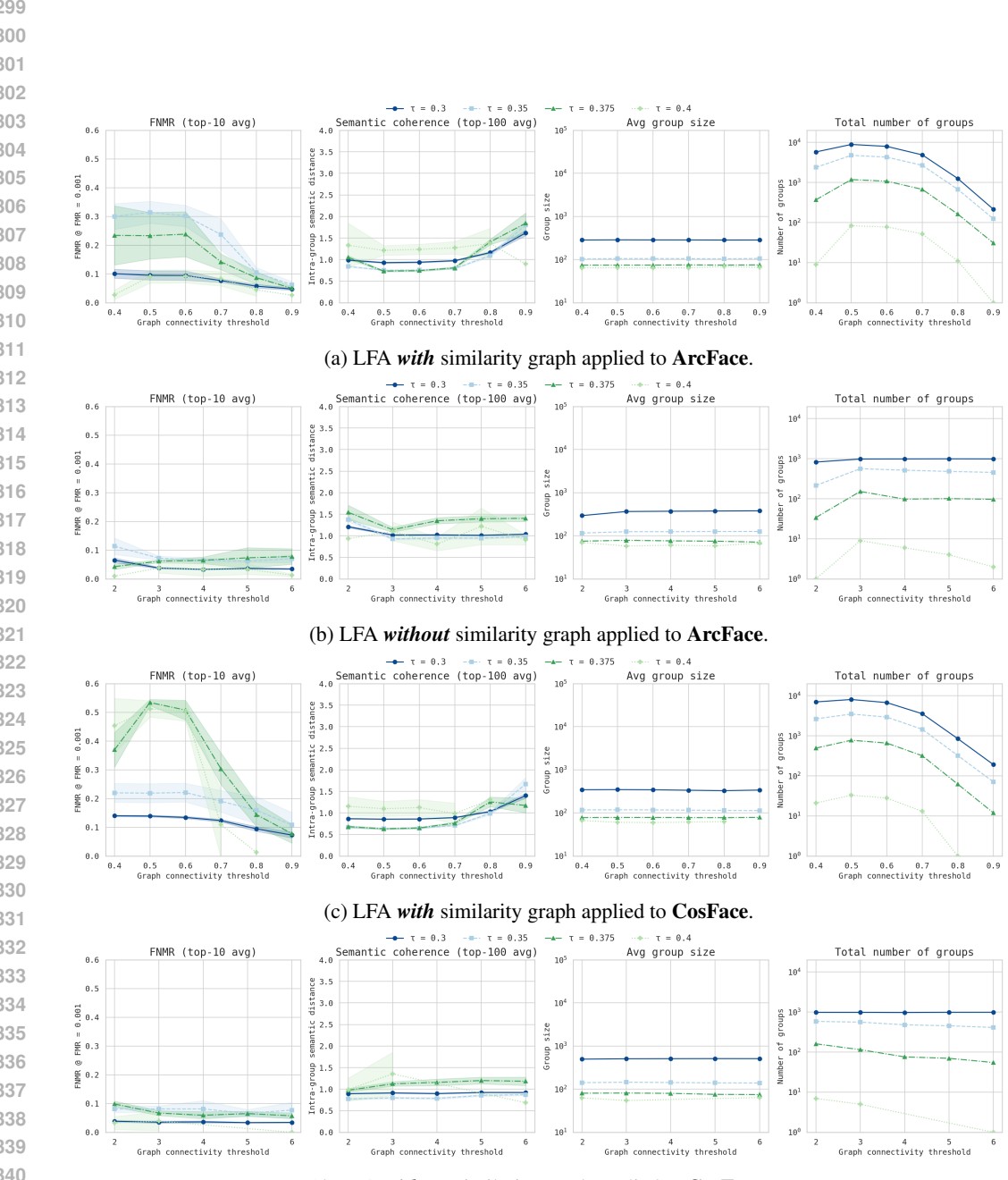

(a) LFA *with* similarity graph applied to **ArcFace**.

(b) LFA *without* similarity graph applied to **ArcFace**.

(c) LFA *with* similarity graph applied to **CosFace**.

(d) LFA *without* similarity graph applied to **CosFace**.

Figure 11: Ablation of the LFA algorithm *with* and *without* similarity graph pre-clustering applied to RFW embeddings obtained from **ArcFace** and **CosFace** models.

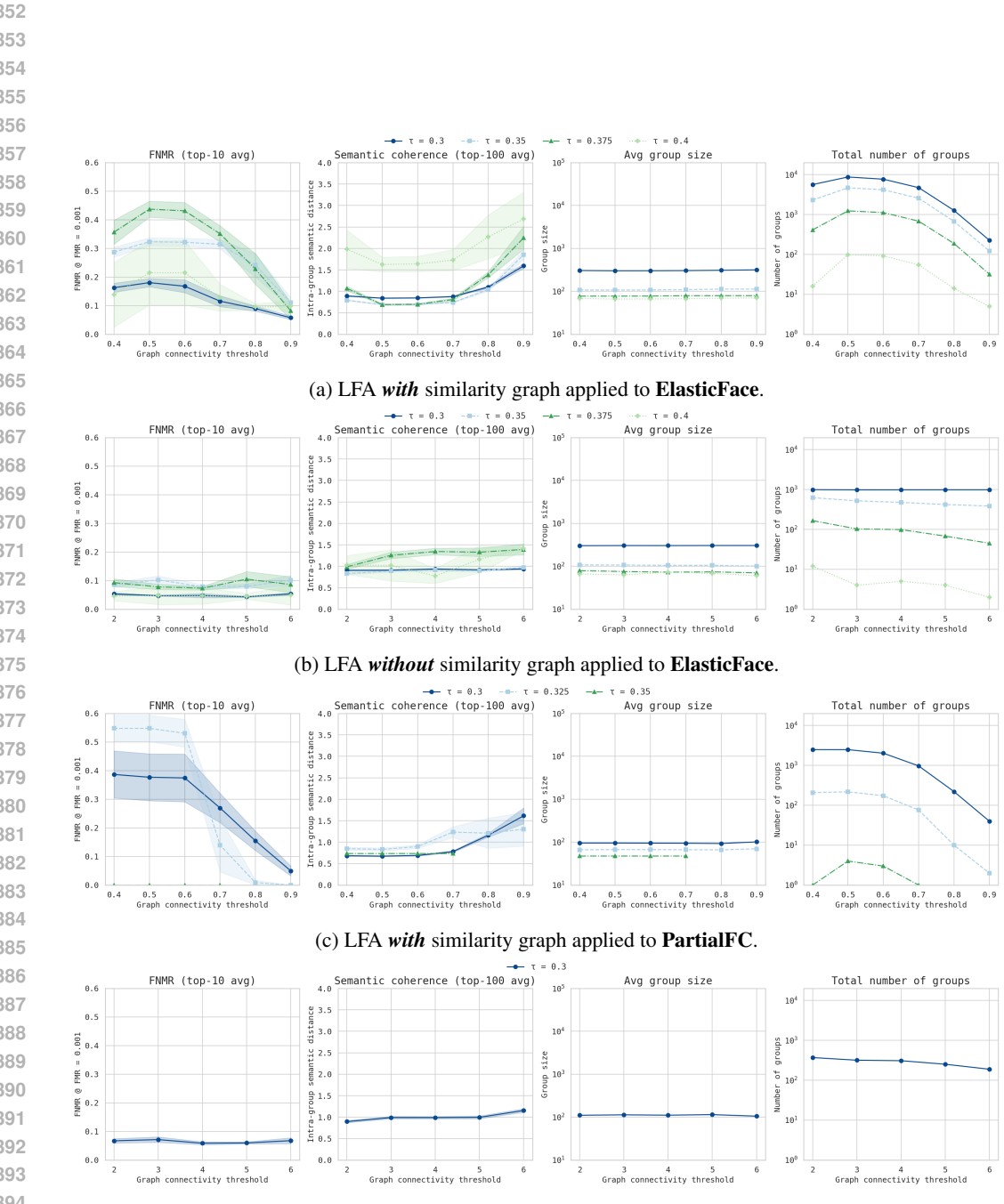

Figure 12: Ablation of the LFA algorithm *with* and *without* similarity graph pre-clustering applied to RFW embeddings obtained from **ElasticFace** and **PartialFC** models.

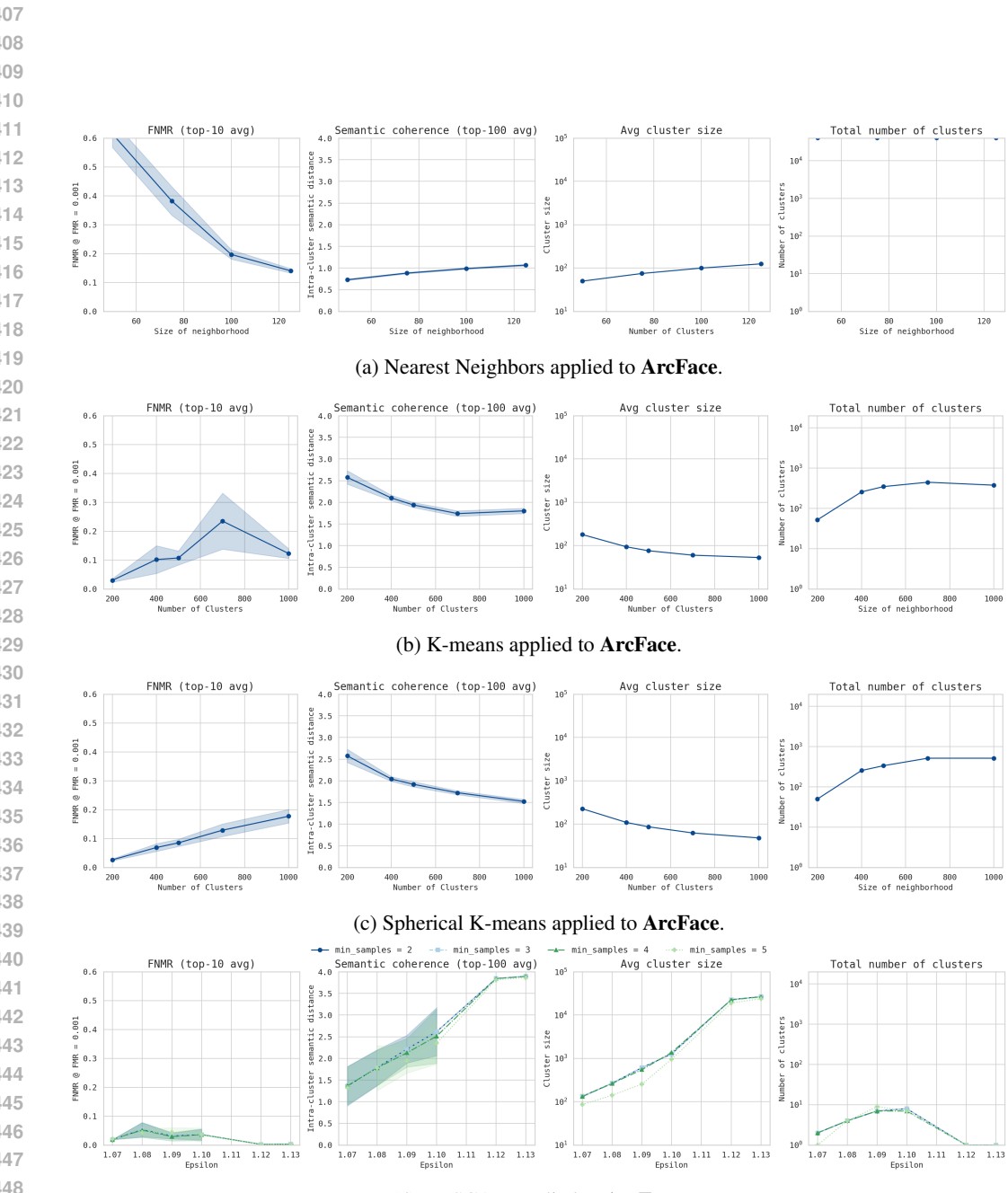

(a) Nearest Neighbors applied to **ArcFace**.

(b) K-means applied to **ArcFace**.

(c) Spherical K-means applied to **ArcFace**.

(d) DBSCAN applied to **ArcFace**.

Figure 13: Ablation of the Nearest Neighbors, k-means, spherical k-means, and DBSCAN algorithms applied to **ArcFace** embeddings from RFW database.

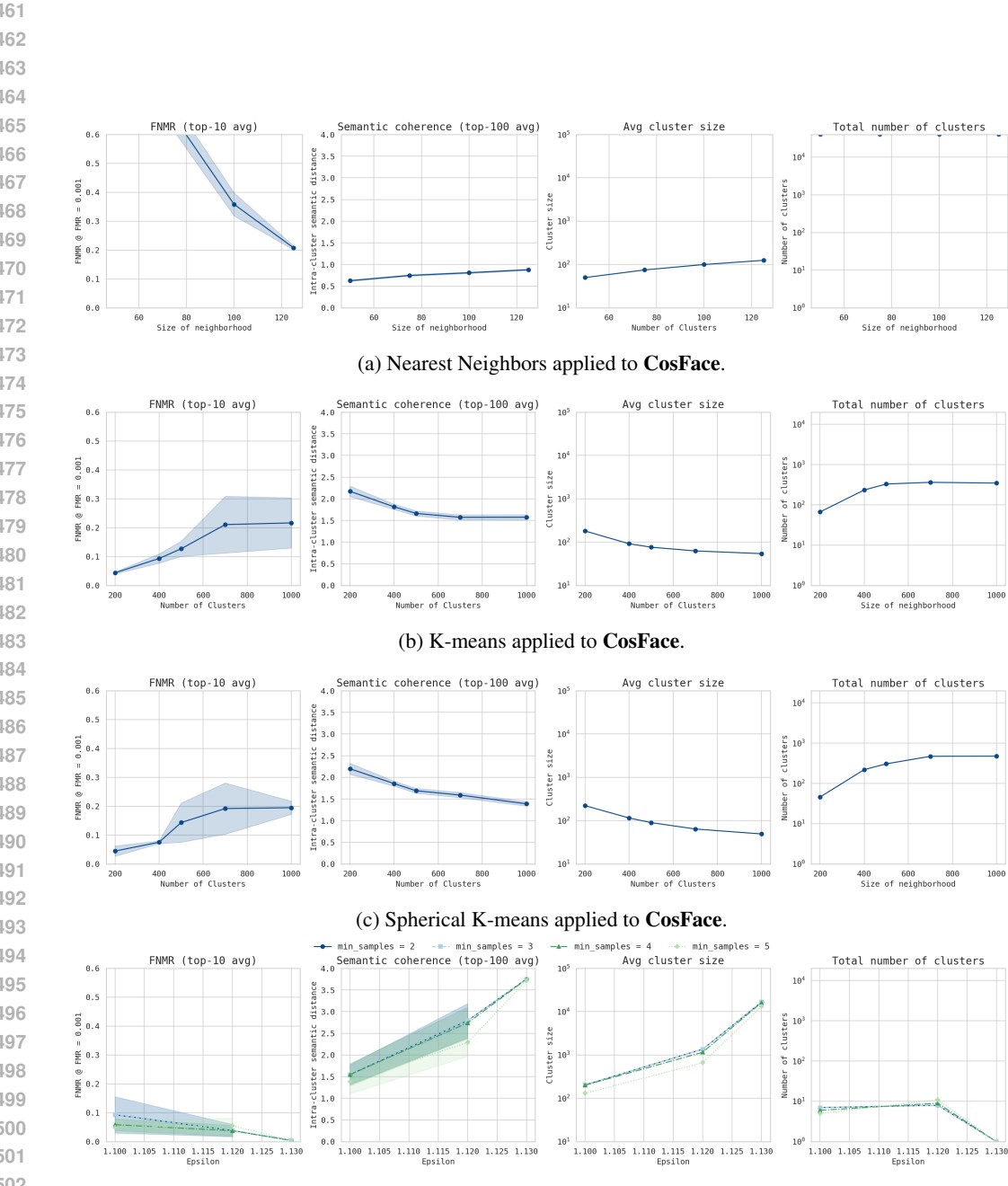

(a) Nearest Neighbors applied to **CosFace**.

(b) K-means applied to **CosFace**.

(c) Spherical K-means applied to **CosFace**.

(d) DBSCAN applied to **CosFace**.

Figure 14: Ablation of the Nearest Neighbors, k-means, spherical k-means, and DBSCAN algorithms applied to CosFace embeddings from RFW database.

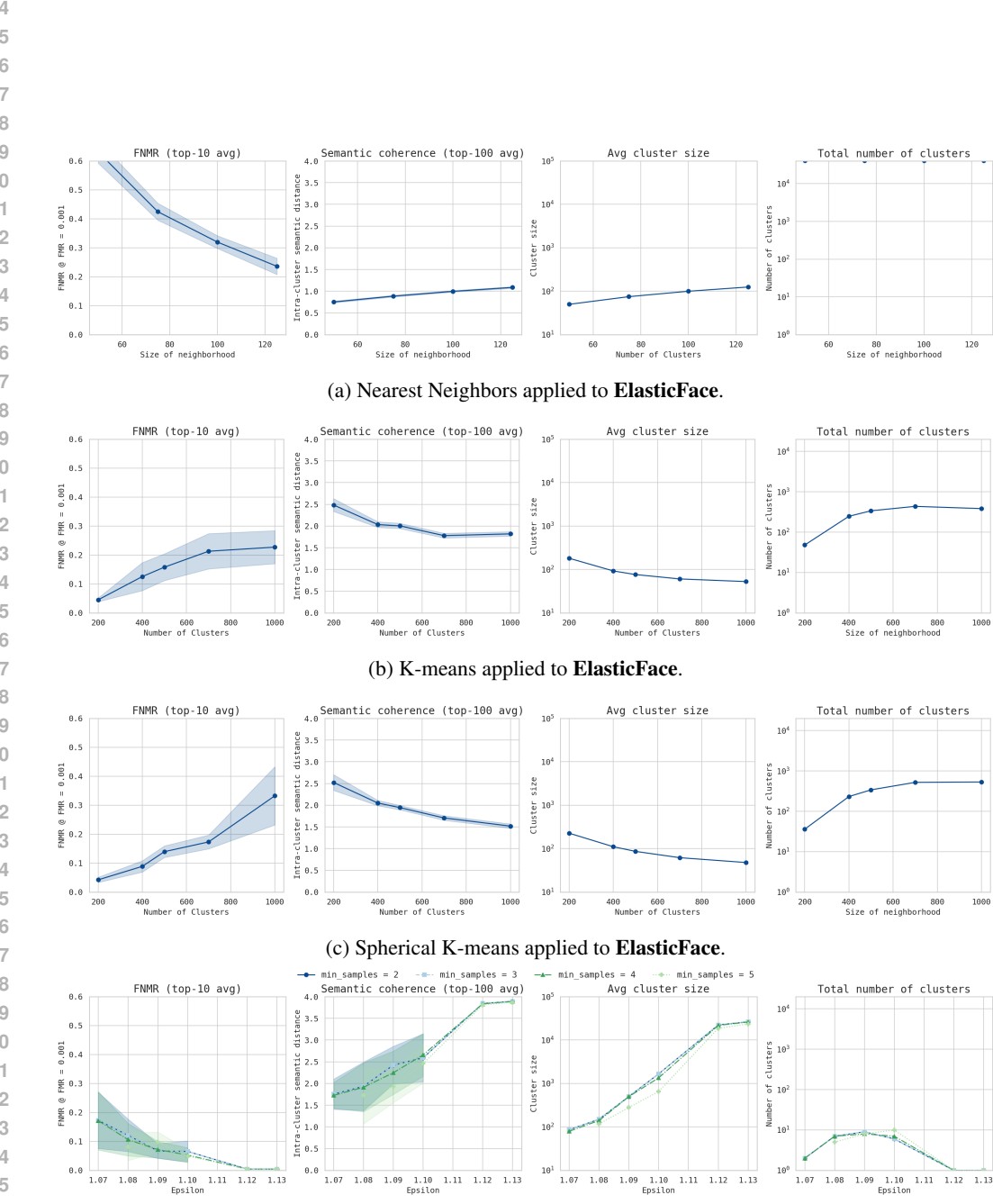

(a) Nearest Neighbors applied to **ElasticFace**.

(b) K-means applied to **ElasticFace**.

(c) Spherical K-means applied to **ElasticFace**.

(d) DBSCAN applied to **ElasticFace**.

Figure 15: Ablation of the Nearest Neighbors, k-means, spherical k-means, and DBSCAN algorithms applied to ElasticFace embeddings from RFW database.

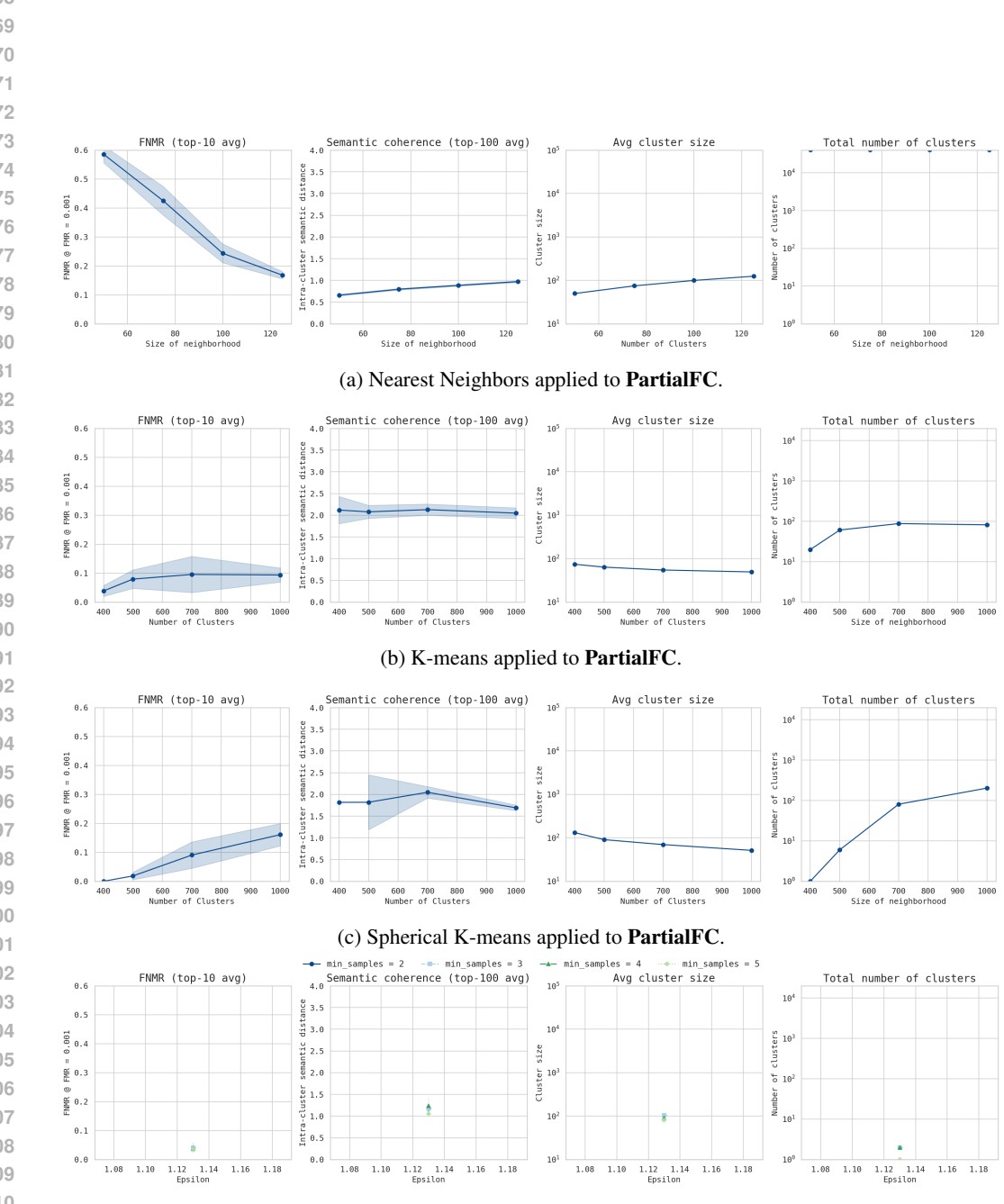

Figure 16: Ablation of the Nearest Neighbors, k-means, spherical k-means, and DBSCAN algorithms applied to PartialFC embeddings from RFW database.

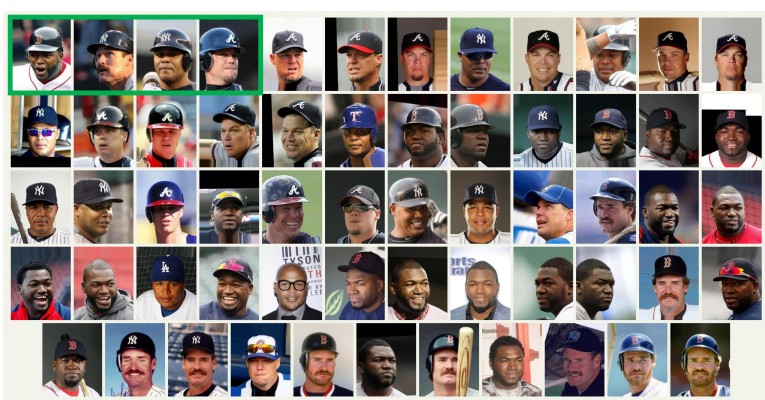

Figure 17: CelebA images of males with baseball helmet. Initial group in green.

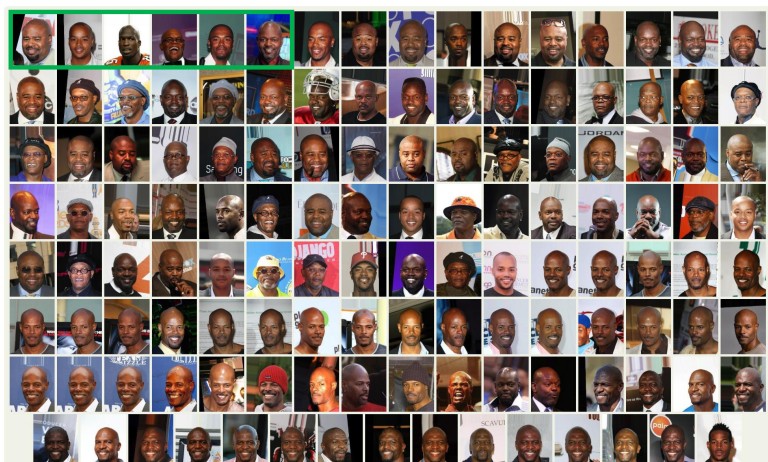

Figure 18: CelebA images of males with black skin and bald. Initial group in green.

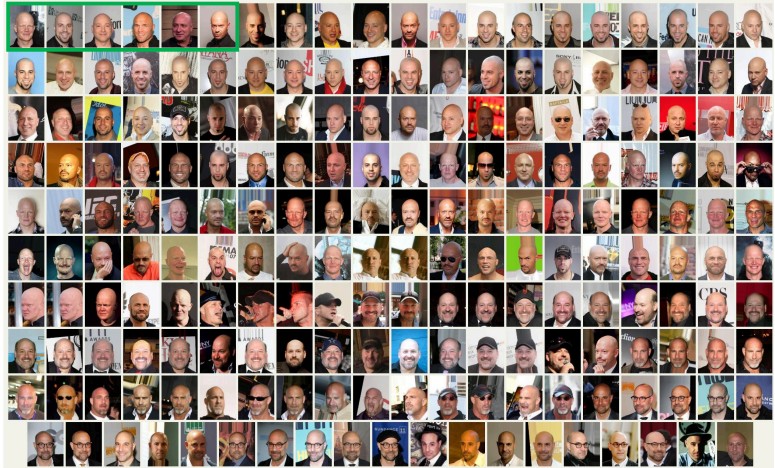

Figure 19: CelebA images of males with white skin and bald. Initial group in green.

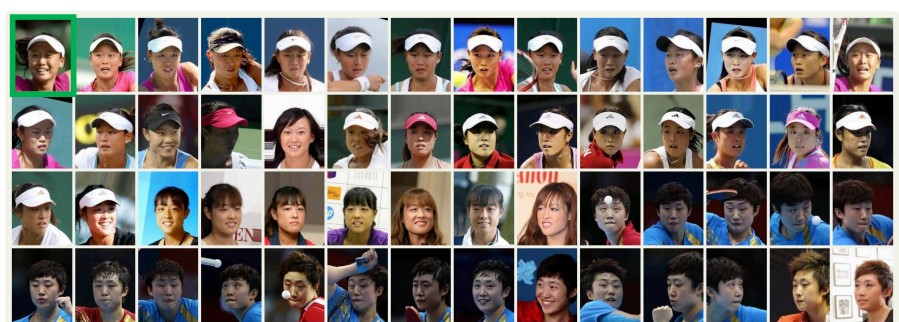

Figure 20: CelebA images of females. Single initial group in green.

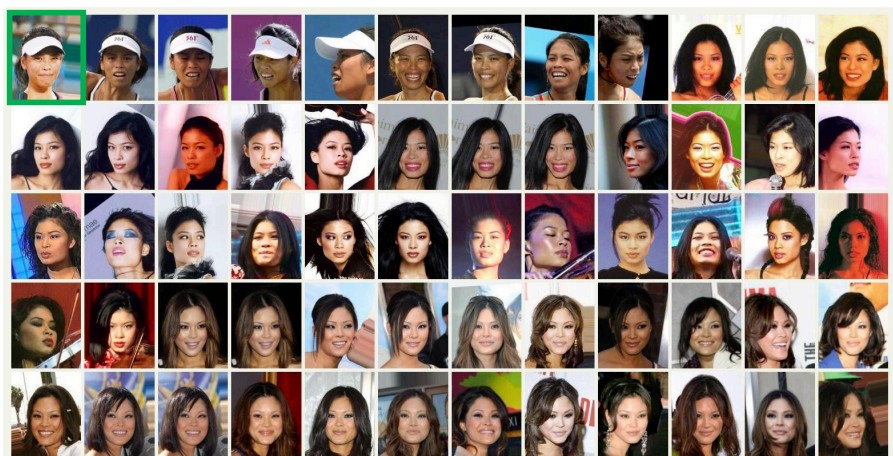

Figure 21: CelebA images of females. Single initial group in green.

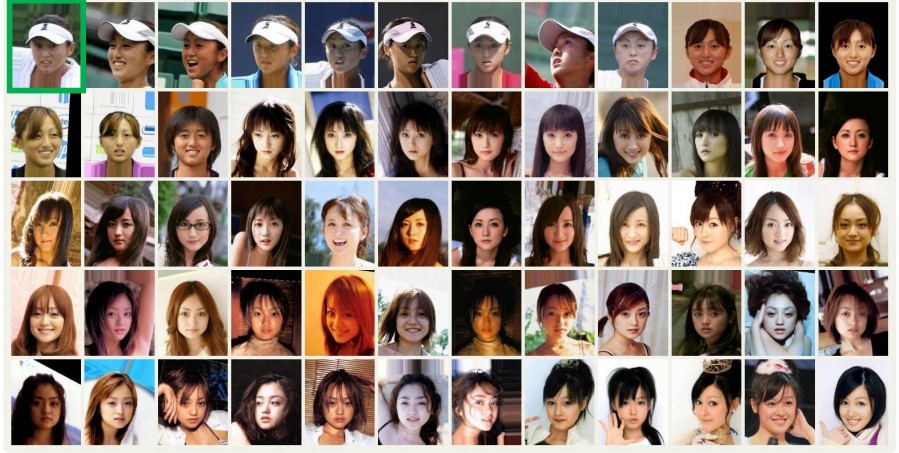

Figure 22: CelebA images of females. Single initial group in green.

## F    IMPLEMENTATION DETAILS

All experiments were conducted on a single NVIDIA A100 GPU with 40GB of memory, an Intel Xeon IceLake-SP 8360Y CPU, and 512GB of RAM.

**Processing Times** Labeling the RFW dataset using visual-language models required several days. Extracting face embeddings with face recognition models took several hours.

Our method was evaluated on two datasets:

RFW (40k images): Runtime ranged from approximately 10 minutes (with a low threshold) to a few hours (with a high threshold).

CelebA (200k images): Runtime ranged from a couple of hours (low threshold) to several hours (high threshold).

