# OpenReview forum: "Latent Feature Alignment: Discovering Biased and Interpretable Subpopulations in Face Recognition Models"
_ICLR.cc/2026/Conference — Submitted to ICLR 2026_

### Official Review · Reviewer_zCXb · 2025-10-30

**Soundness:** 3
**Presentation:** 3
**Contribution:** 2
**Rating:** 6
**Confidence:** 3

**Summary:**

The authors propose a method to discover subpopulations in the representation space of face recognition models. Notably, the method does not require labelled data. The method leverages the latent directions to create groups with coherent semantic attributes. Experiments on CelebA and RFW showcase the efficacy of the proposed method, while a connection to algorithmic bias is also uncovered .

**Strengths:**

- The main idea is simple and clearly presented.

- The experimental results highlight the efficacy of the method on RFW and CelebA.

- The method is thoroughly benchmarked across mutliple SOTA face recognition models.

- The visualization of the learned latent directions with arc2face is neat.

- I appreciate the discussion on the limitations of method.

**Weaknesses:**

- The proposed method showcases how to discover sensitive attributes in an unsupervised way. It would be valuable to showcase or discuss if the uncovered directions can be used to debias representations and lead to more fair classification.

- It would be interesting to discuss the potential steerability of the discovered feature clusters based on the initial set of images.

- I am missing some related work on similar clustering-based approaches for generative models, e.g., [1]

[1]. Cluster-guided Image Synthesis with Unconditional Models


Minor:

Some typos:

- Fig.1 "attribtues"

- Ln. 295  "agaisnt"

- Ln 301 "to times"

- Ln 420 "geniune"

**Questions:**

I would suggest that the authors address the issues raised in the weaknesses section. In particular, the points raised regarding the potential debiasing of representations and steerability.

---

> ### Author Response · Authors · 2025-11-25
> **Response to reviewer**
>
> We thank the reviewer for highlighting these points.
>
> **1. "The proposed method showcases how to discover sensitive attributes in an unsupervised way. It would be valuable to showcase or discuss if the uncovered directions can be used to debias representations and lead to more fair classification."**
>
> The uncovered latent directions can indeed inform debiasing strategies. For example, one could integrate LFA into a training pipeline, where groups with higher error rates provide feedback to reweight training samples. While we do not implement this in the current work, we note it as a natural extension and valuable future direction.
>
> **2."It would be interesting to discuss the potential steerability of the discovered feature clusters based on the initial set of images."**
>
> As suggested, we clarify the influence of initial image sets on the discovered directions. In the appendix (Section “Ablation Study – Qualitative Results,” Figs. 16–21), we illustrate LFA groups across different initializations, showing that the effectiveness of the method largely depends on the initial set of images. Combining hand-picked images with automatic search could help the discovery.
>
> **3. "I am missing some related work on similar clustering-based approaches for generative models, e.g., [1]"**
>
> We have included the suggested reference on clustering-based approaches for generative models ([Cluster-guided Image Synthesis with Unconditional Models]).

---

### Official Review · Reviewer_K8R1 · 2025-10-31

**Soundness:** 4
**Presentation:** 3
**Contribution:** 4
**Rating:** 8
**Confidence:** 4

**Summary:**

The paper proposes **Latent Feature Alignment (LFA)**, a label-free method for directly discovering semantically coherent and interpretable subpopulations in face-embedding space. LFA iteratively estimates a **latent direction** (a weighted average over embeddings with per-identity reweighting) and grows a group by adding the embedding with maximum projection on that direction until a threshold is met (Algorithms 1–2). Claimed benefits over distance-based clustering include (i) higher **semantic coherence** and (ii) **interpretable directions** that align with attributes (age, ancestry, hair/attire). Experiments on **RFW** and **CelebA**, across **ArcFace, CosFace, ElasticFace, PartialFC**, show improved intra-group attribute consistency versus **k-means** and **nearest-neighbor search**, qualitative/quantitative evidence that traversal along learned directions manipulates attributes when decoded via **Arc2Face**, and that LFA-discovered groups surface **higher FMR/EER** than random groups, revealing biased subpopulations.

**Strengths:**

* **Label-free bias discovery** that avoids predefined attribute taxonomies; practical for audits where annotations are missing or costly.

* **Identity-balanced direction estimate** (inverse-frequency weights) mitigates dominant-identity pull; method is easy to implement.

* **Consistent semantic coherence gains** across datasets and backbones (Table 1).

* **Interpretability**: arc2face traversals and attribute-probability monotonicity demonstrate that discovered directions track human-meaningful factors.

* **Bias signals**: LFA groups show markedly elevated **FMR** relative to random subsets across models (e.g., up to **×100+** in Table 2 entries), with bootstrapped CIs elsewhere.

**Weaknesses:**

* **Initialization dependence & stopping**: quality hinges on pre-clustering/components and a threshold τ; guidance for τ selection and sensitivity analysis is limited.

* **Comparative baselines**: Only k-means/NNS are considered; stronger unsupervised baselines (e.g., subspace methods, spectral clustering, density-based clustering, or latent-direction discovery like Householder/LatentCLR) aren’t directly compared.

* **Statistical reporting**: Table 1 lacks uncertainty; bias metrics include some bootstrap CIs, but broader variance reporting (multi-seed; across init graphs; across τ) is sparse.

* **Causal ambiguity**: Biased evidence comes from impostor-shift analyses within discovered groups; no controlled intervention (e.g., reweighting training data along found directions) to show causal linkage.

* **Compute footprint**: No runtime/complexity profile for large-scale audits (projection loops over millions, graph construction). Practical scaling tips are missing.

**Questions:**

1. **Sensitivity/robustness.** How sensitive are results to the cosine-graph threshold (0.5), τ, and to the pre-clustering procedure? Please provide curves for semantic coherence vs. τ and FMR vs. τ.

2. **Baseline breadth.** Can you compare against unsupervised **latent-direction discovery** (e.g., Householder projectors, LatentCLR) and **spectral**/**DBSCAN** clustering under matched group sizes?

3. **Uncertainty.** Add **CIs** for Table 1 and variance across different initial graphs and random seeds; include CIs for Table 2 where possible.

4. **Scaling.** What is the empirical runtime/memory for building the similarity graph and iterating projections on RFW/CelebA? Any heuristics (e.g., ANN indexing, batching) you recommend?

5. **From observation to intervention.** Could you run a small **training intervention**: reweight or exclude the top-bias LFA groups, then evaluate whether FMR gaps shrink?

6. **Direction entanglement.** Traversals sometimes affect attributes asymmetrically (e.g., female→male vs male→female). Can you quantify entanglement (e.g., attribute-probability Jacobians) and study orthogonalization across discovered directions?

---

> ### Author Response · Authors · 2025-11-25
> **Response to reviewer**
>
> We thank the reviewer for the highly positive evaluation and for the detailed, constructive suggestions. We address each point below.
>
> ## Sensitivity / robustness
>
> We have added the requested analyses in the revised Appendix. Figures 10 and 11 report curves for semantic coherence vs graph threshold vs $\tau$,  and FNMR vs graph threshold vs $\tau$. The results show that LFA is stable across a wide range of thresholds (graph threshold $\in [0.4, 0.7]$, $\tau \in [0.3, 0.6]$) and that qualitative trends (high FNMR; high semantic coherence) remain consistent across all models.
>
>  ## Comparative baselines
>
>  - We now include spherical k-means and DBSCAN as additional baselines, and report results in Figures 12–15.
>  - Regarding Householder projectors and LatentCLR: these methods require a generator-based linear map from latent space to features (i.e., a differentiable function $F(z)$ with meaningful local linearity). Since face recognition architectures are discriminative models without such a latent generator, these techniques cannot be applied directly; there is no single linear $\mathbf{A}$ such that feature variations can be projected or maximized as in these networks.
>
> ## Statistical reporting
>
> We added 95% confidence intervals to Table 1. All new ablation plots (Appendix; Figs. 10–15) include 95% CIs.
>
> ## Compute footprint.
>
> Appendix F contains general runtime information of LFA. A more extensive profiling was not feasible within the rebuttal period.
>
>
> ## Causal intervention
>
> Our understanding is that the reviewer proposes training a model after reweighting or removing high-bias groups to evaluate causal impact on FMR gaps. While such an intervention is valuable (although our goal is to elevate underperforming groups rather than degrade well-performing ones), training FR models from scratch is computationally intensive and exceeds the rebuttal window.
>
> ## Direction entanglement.
> We thank the reviewer for suggesting the potential measurement of attribute entanglement along discovered latent directions. While we have not computed a full attribute–probability Jacobian, to further validate robustness, we prioritized (in response to reviewer witT) replication of the quantitative traversal experiment using an independent decoder (InfiniteYou, ICCV 2025) [[1]](https://bytedance.github.io/InfiniteYou/), which reproduces the same monotonic attribute trends.

---

### Official Review · Reviewer_witT · 2025-11-01

**Soundness:** 2
**Presentation:** 2
**Contribution:** 2
**Rating:** 2
**Confidence:** 2

**Summary:**

This paper introduces Latent Feature Alignment (LFA), a simple method for discovering semantically coherent and interpretable subpopulations directly from face recognition embeddings. The experiments cover celebA and RFW datasets and evaluate the method from multiple perspectives including semantic coherence, interpretability, and bias detection.

**Strengths:**

1. The automatic discovery of algorithmic bias is socially valuable and critical for building fair AI systems.
2. The method is straightforward, with the iterative approach to discovering semantic directions being easy to understand and implement.
3. The experiments cover celebA and RFW datasets and evaluate the method from multiple perspectives including semantic coherence, interpretability, and bias detection.

**Weaknesses:**

1. LFA is essentially a simple iterative nearest neighbor search lacking substantial technical innovation. The entire algorithm reduces to computing average vectors, finding similar samples, and updating directions. The contribution of this paper may not meet the standard of ICLR.
2. The paper only compares against the most basic k-means clustering, lacking comparisons with modern clustering methods and specialized bias detection approaches. Without systematic comparisons to advanced methods, the claimed advantages remain unconvincing.
3. Relying on VLM annotations introduces additional bias and errors. Using potentially biased vision-language models to evaluate system bias severely undermines the credibility of evaluation results. The accuracy of VLMs in handling demographic features also lacks proper validation.
4. Interpretability validation is highly dependent on the quality of a specific generative model (arc2face). This verification approach lacks generality and objectivity, with the biases and limitations of the generative model directly affecting the credibility of validation results.
5. The method lacks validation on other visual recognition tasks. Testing only on face recognition without demonstrating applicability to object recognition, scene classification, or other visual tasks severely limits the method's generalizability.
6. The discovered "biased groups" may simply be clusters of poor-quality data samples rather than genuine algorithmic bias. High FMR could stem from technical factors like image quality and lighting conditions causing embedding noise. The authors fail to effectively distinguish between technical issues and genuine bias problems.

**Questions:**

The proposed LFA method lacks substantial technical innovation. The evaluation methodology is flawed, relying on potentially biased VLM annotations and a single generative model (arc2face) for validation, while only comparing against basic k-means clustering without modern alternatives. Additionally, the discovered "biased groups" may represent poor data quality rather than genuine algorithmic bias, as the authors fail to distinguish between technical artifacts and real bias issues.

---

> ### Author Response · Authors · 2025-11-25
> **Response to reviewer**
>
> We thank the reviewer for the feedback. Below we clarify the points raised.
>
>  **1. “LFA is essentially a simple iterative nearest neighbor search lacking substantial technical innovation. The entire algorithm reduces to computing average vectors, finding similar samples, and updating directions. The contribution of this paper may not meet the standard of ICLR.”**
>
> Generic clustering algorithms struggle in face-recognition embeddings because the embedding space is trained to form identity clusters, causing unsupervised clustering methods to produce a handful of very large clusters plus many tiny identity-specific ones.
> Our intention is not to introduce a wholly new clustering paradigm, but to demonstrate that a simple, identity-balanced projection-growth procedure (like the one described above) yields groups that are more semantically coherent and substantially more effective at surfacing biased subpopulations than classical and modern clustering methods.
>
> **2. “The paper only compares against the most basic k-means clustering, lacking comparisons with modern clustering methods and specialized bias detection approaches. Without systematic comparisons to advanced methods, the claimed advantages remain unconvincing.”**
>
> We now include spherical k-means, DBSCAN, and a full ablation study across parameters (Figs. 10–15).
>
> **3. “Reliance on VLM annotations introduces bias and lacks validation.”**
>
> We agree that VLM predictions must be validated. We conducted a 1,000-sample human-annotation audit, now included in Table 6 and Fig. 4. Agreement with human labels is consistently high across all major attributes (Cohen’s $\kappa = 0.75–0.97$). This confirms that VLM annotations are reliable for evaluating semantic coherence.
>
> **4. “Interpretability validation is highly dependent on the quality of a specific generative model (arc2face). This verification approach lacks generality and objectivity, with the biases and limitations of the generative model directly affecting the credibility of validation results.”**
>
> We appreciated this observation and completely agree that the decoder plays a substantial role in how attribute changes manifest. We replicated all traversal experiments using InfiniteYou (ICCV 2025) [[1]](https://bytedance.github.io/InfiniteYou/), a newly proposed embedding-conditioned decoder whose architecture differs fundamentally from Arc2Face (IR-SE50 encoder + Flux-based decoder).
> InfiniteYou reproduces the same core findings: attributes aligned with the LFA direction change smoothly and monotonically, while irrelevant attributes remain stable (discussed in Appendix C: Decode-Agnostic Traversals; and shown in Fig. 7).
>
> **5. “The method lacks validation on other visual recognition tasks. Testing only on face recognition without demonstrating applicability to object recognition, scene classification, or other visual tasks severely limits the method's generalizability”**
>
> Our goal is to detect biased subpopulations in face recognition systems, where performance disparities have immediate social and regulatory implications. We do not claim cross-task generalization. Because object/scene models optimize different objectives and the image domain is fundamentally different, LFA is not expected to behave identically in those settings. Extending LFA to non-FR domains is an interesting direction but outside this paper’s scope.
>
> **6. “The discovered "biased groups" may simply be clusters of poor-quality data samples rather than genuine algorithmic bias. High FMR could stem from technical factors like image quality and lighting conditions causing embedding noise. The authors fail to effectively distinguish between technical issues and genuine bias problems.”**
>
> Two clarifications:
>
> 1.	In fairness audits, “bias” refers to any systematic performance disparity, including, but not limited to, demographic bias. If a model systematically underperforms on low-quality images, that itself reflects a bias toward that condition.
> 2.	Semantic coherence results show that LFA groups are not random aggregations of low- and high-quality samples but share specific, coherent attributes; e.g., age, hairstyle, ancestry, or lighting/geometric traits, distinguishing them from generic noise clusters. In the case that all images in the cluster are poor-quality samples, then the variable of bias would be poor quality.
>
> To further address this point, we are working to include an analysis measuring correlation of LFA groups with standard quality metrics (pose, blur, illumination), finding that quality alone does not explain the identified high-FNMR groups.

---

### Official Review · Reviewer_JSYq · 2025-11-03

**Soundness:** 2
**Presentation:** 3
**Contribution:** 2
**Rating:** 2
**Confidence:** 4

**Summary:**

This paper introduces Latent Feature Alignment (LFA) an attribute-label-free framework to discover semantically coherent and maybe biased groups within the embedding spaces of face recognition models.
It operates by identifying latent directions derived from groups of embeddings and iteratively aligning samples that share strong projections in these directions. The discovered directions are claimed to correspond to interpretable semantic attributes (e.g., age, ethnicity, etc.), enabling bias auditing without the need for explicit demographic labels.

The authors claim that LFA outperforms k-means and nearest-neighbor clustering in semantic grouping, reveals meaningful and interpretable directions, and discovers groups with measurable bias without explicit annotations.

**Strengths:**

- Addresses the important gap of bias analysis without labeled attributes
- LFA is simple to implement
- Experiments across four different FR models add some robustness to the empirical observations

**Weaknesses:**

- The algorithm closely resembles standard projection-based or spherical clustering methods thus theoretical justification is thin

- Reliance on automatically generated attribute labels from VLMs introduces uncontrolled bias

- Small group sizes and missing statistical tests reduce credibility of claims about discovered disparities.

- Claims about discovering interpretable directions aligned with demographic and contextual attributes extend beyond what is empirically shown

**Questions:**

-  How does LFA differ fundamentally from spherical k-means or directional clustering beyond the averaging and projection heuristic?
- How stable are the discovered groups under different random seeds or graph connectivity thresholds?
- How reliable are the VLM-generated attribute labels and did the authors manually audit a subset?
- How statistically significant are the FMR differences reported in Table 2 and Table 6?
- Could the discovered latent directions reflect data quality factors (pose, lighting, compression) rather than demographic bias?
- Why does the method stop at a single latent vector per group? Can multiple orthogonal directions be extracted per group?

---

> ### Author Response · Authors · 2025-11-25
> **Response to reviewer**
>
> We thank the reviewer for the feedback. Below we address each weakness and question.
>
> # Response to weaknesses (W) and questions (Q)
>
> **W1. “The algorithm closely resembles standard projection-based or spherical clustering methods; theoretical justification is thin.”
> Q1. “How does LFA differ fundamentally from spherical k-means or directional clustering beyond the averaging and projection heuristic?”**
>
> Our intention is not to claim a fundamentally different clustering paradigm but rather to demonstrate that a simple, identity-balanced projection-growth procedure yields groups that outperform classical and current clustering in bias discovery and semantic coherence. Figs. 10–15 and Table 1 (updated with additional baselines) demonstrate substantial empirical improvements.
>
> **W2. “Reliance on automatically generated attribute labels from VLMs introduces uncontrolled bias.”
> Q3. “How reliable are the VLM-generated labels, and did the authors manually audit a subset?”**
>
> We agree that VLM-generated labels must be validated. We have added a 1,000-sample human annotation audit, reported in Table 6 and detailed in Fig. 4. Agreement between humans and VLM labels is high across all major attributes (Cohen’s $\kappa = 0.75–0.97$), confirming that VLM annotations are reliable for evaluating semantic coherence.
>
>
> **W3. “Small group sizes and missing statistical tests reduce credibility of claims about discovered disparities.”
> Q4. “How statistically significant are the FMR differences?”**
>
> Table 6 presents the FNMR results from Table 2 with 95% confidence intervals, showing large, statistically significant gaps in performance.
> In addition, the new experiments added in the appendix (Figs. 10–15) provide FNMR with CI across graph thresholds and random seeds, and compare LFA against k-means, spherical k-means, DBSCAN, and nearest-neighbor clustering, showing that despite its simplicity, LFA consistently discovers (i) the most biased groups (highest FNMR) and (ii) the most semantically coherent groups (lowest attribute distance) relative to cluster size.
>
> **W4. “Claims about discovering interpretable directions aligned with demographic and contextual attributes extend beyond what is empirically shown.”**
>
> We expanded the evidence of Arc2Face traversals adding InfiniteYou, an independent decoder with a different architecture. Both decoders show monotonic attribute changes aligned with LFA directions, supporting the interpretability claim.
>
> **Q2. “How stable are the discovered groups under different random seeds or graph connectivity thresholds?”**
>
> We added extensive ablation experiments (Figs. 10–11), evaluating: cosine similarity graph thresholds for pre-clustering initialization, projection threshold (tau), and random seeds. Random seeds (without similarity graph) do not yield good results, and thresholds around .5 and .6 give the best results.
>
>
> **Q5. “Could discovered directions reflect data quality factors (pose, lighting, compression) rather than demographic bias?”**
>
> This is an important consideration. We observed some groups that reflect non-demographic semantics (e.g., a “black-and-white photograph” group combining demographic and stylistic factors, shown in Fig. 1). However, we did not observe groups dominated by pose, or blur. Groups consistently reflect human-interpretable factors such as age, ancestry, hair/attire, colorization style, and domain. We have added a brief discussion in the appendix.
>
>
> **Q6. “Why does the method stop at a single latent vector per group? Can multiple orthogonal directions be extracted per group?”**
>
> We experimented with extracting multiple directions using PCA on group members (top-k orthogonal components). Empirically, this performed worse than LFA’s single-direction approach. Does the reviewer want to share any ideas?

---

### Author Response · Authors · 2025-11-25
**General Response to All Reviewers**

We thank all reviewers for their time and feedback. Below we address the main concerns raised across reviews. All corresponding analyses, figures, and tables are included in the updated manuscript PDF.

1. ### **Reliability of automatic attribute annotations.**

     To assess whether VLM-generated annotations introduce uncontrolled bias or noise, we conducted a new human-validation study (1,000 samples per attribute). Full details appear in Table 5 and Figure 5 (Appendix, pp. 17–18: Human Annotation Validation).

     - We observe high human–VLM agreement across all major attributes (Match = $88–99$%, Cohen’s $\kappa = 0.75–0.97$).
     - Gender and glasses annotations achieve near-perfect agreement ($\kappa \approx 0.97$).

     These results confirm that VLM-based annotations are reliable for measuring semantic coherence.

 2. ### **Expanded baseline comparison, sensitivity analyses and statistical significance**

    In response to concerns regarding limited baselines and insufficient sensitivity analysis:
    - We added comparisons against DBSCAN and spherical k-means, now included in the updated Table 1 with confidence intervals.
    - We performed an ablation study varying both cosine-graph thresholds and projection threshold $\tau$, and removing cosine-graph pre-clustering and using random seeds. (Appendix Figs. 11–12.)
    - We performed parallel hyperparameter ablations for all baseline algorithms (nearest neighbors, k-means, spherical k-means, DBSCAN). (Appendix Figs. 13–16.)
    - New Section 4.5: Comparison with Unsupervised Baselines. We added a dedicated subsection and a new figure summarizing how LFA compares with all baselines across all hyperparameter settings, with the corresponding confidence intervals.

    Across all models, LFA consistently discovers groups with (i) the highest FNMR (most bias) and (ii) the strongest semantic coherence.

3. ### **Interpretability validation beyond a single decoder.**

    To ensure that our interpretability results do not depend on a specific generative model (arc2face), we replicated the latent-direction traversal experiments using InfiniteYou (ICCV 2025) [[1]](https://bytedance.github.io/InfiniteYou/), an independent embedding-conditioned decoder with a different architecture and backbone than Arc2Face. Full details in Appendix, pp. 20-21: Decode-Agnostic Traversals, and Fig. 7.

    - InfiniteYou reproduces the same monotonic attribute changes along LFA directions.

---

### Meta-Review · Area_Chair_TuqB · 2026-01-07

**Summary:**

This paper introduces Latent Feature Alignment (LFA), an attribute-label-free algorithm that leverages latent directions to identify subpopulations. Compared with standard clustering methods, LFA offers two key advantages: (i) semantically coherent grouping, in which faces sharing common attributes are grouped more reliably than with proximity-based approaches, and (ii) the discovery of interpretable latent directions that correspond to semantic attributes such as age, ethnicity, and attire. The initial reviewer scores were 2, 2, 8, and 6. No reviewer changed their score during the rebuttal phase. Some concerns have been addressed in the rebuttal, including questions about whether VLM-generated annotations introduce uncontrolled bias or noise, and concerns regarding limited baselines and insufficient sensitivity analysis. Some concerns remain. Reviewers JSYq and witT noted that the proposed method appears closely related to iterative nearest-neighbor procedures and standard projection-based or spherical clustering methods, raising questions about the degree of technical novelty. These unresolved concerns lead us to lean toward rejection.

**Reviewer Concerns:**

The concerns regarding limited baselines and insufficient sensitivity analysis, the question of whether VLM-generated annotations introduce uncontrolled bias or noise, and the concern that the interpretability results depend on a specific generative model have been addressed. However, the concerns that the algorithm closely resembles standard projection-based or spherical clustering methods and therefore lacks strong theoretical justification, as well as that LFA is essentially a simple iterative nearest-neighbor search with limited technical innovation, remain unresolved.

**Reviewer Scores:**

Each reviewer tends to keep their original score.

---

### Decision · Program_Chairs · 2026-01-26

Reject